# Hyperbolic Multimodal Continual Learning

**Jiahong Liu**[1]  **Ming Shen**[1]  **Xiaohao Liu**[2]  **Rex Ying**[3]  **Menglin Yang**[4]  **Tat-Seng Chua**[2]  **Irwin King**[1]

## Abstract

Hyperbolic geometry has recently emerged as a powerful representation space for multimodal learning, as it naturally captures hierarchical semantic structure across modalities. Despite this progress, how such representations behave under continual learning poses fundamentally different challenges that remain underexplored. This work provides a geometric perspective on this problem and establishes a theoretical foundation for representation preservation in hyperbolic space, showing that preventing forgetting requires cross-modal invariance under a shared hyperbolic isometry. We further show that forgetting in hyperbolic continual learning involves both semantic relation drift and hierarchy-related distortion, motivating preservation of both cross-modal relational structure and hierarchical geometry. Guided by these insights, a principled continual learning framework is derived that preserves essential geometric structure while allowing effective adaptation to new tasks. Experiments on continual multimodal benchmarks corroborate the effectiveness of the proposed approach. The code is available in the project repository.

## 1. Introduction

Multimodal Contrastive Learning (MCL) (Radford et al., 2021; Dufumier et al., 2025; Liu et al., 2025c;b) has become a dominant paradigm for learning joint representations across modalities, enabling effective alignment between heterogeneous data sources. By leveraging large-scale multi-

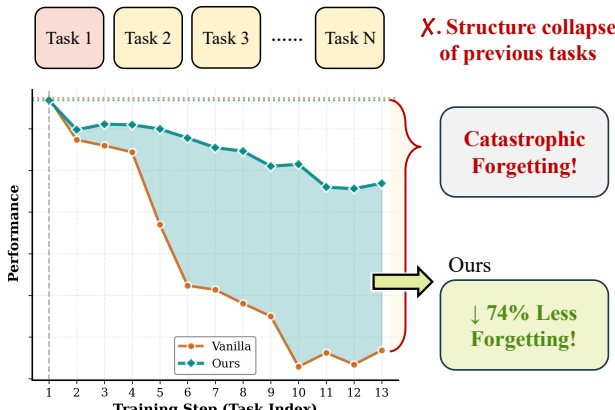

**Continual Learning of Multimodal Hyperbolic Representations**

*Figure 1.* **Continual learning in hyperbolic representation space.** Multimodal representations are embedded in hyperbolic space to capture semantic hierarchy across tasks. Under naive continual updates, distortions in the hyperbolic geometry lead to catastrophic forgetting and collapse of previously learned structures, whereas structure-aware hyperbolic updates significantly reduce forgetting.

modal data and contrastive objectives, such models exhibit robust zero-shot transfer capabilities within a fixed representation space (Radford et al., 2021; Han et al., 2023; Chen et al., 2023; Li et al., 2025). Most existing approaches embed multimodal representations in high-dimensional Euclidean spaces, implicitly assuming a flat geometric structure for semantic concepts (Radford et al., 2021; Han et al., 2023; Chen et al., 2023).

However, multimodal semantic concepts often exhibit inherent hierarchical structure (Desai et al., 2023; Pal et al., 2025; Kim et al., 2024). Across modalities, abstract and high-level concepts typically subsume a large number of more specific instances, inducing asymmetric and partially ordered semantic structures rather than flat metric clusters. Such differences in semantic specificity and abstraction are difficult to faithfully capture in Euclidean spaces, where distances and neighborhoods are uniformly structured (Krioukov et al., 2010; Nickel & Kiela, 2017; 2018; Ganea et al., 2018a).

**Hyperbolic Multimodal Learning.** Hyperbolic geometry provides a principled representation space for hierarchical semantics (Krioukov et al., 2010; Nickel & Kiela, 2017; 2018; Ganea et al., 2018b;a). Recent advances in hyperbolic

[1]Department of Computer Science and Engineering, The Chinese University of Hong Kong, Hong Kong, China [2]School of Computing, National University of Singapore, Singapore [3]Department of Computer Science, Yale University, New Haven, CT, United States [4]AI Thrust, The Hong Kong University of Science and Technology (Guangzhou), Guangzhou, Guangdong, China. Correspondence to: Jiahong Liu <jiahong.liu21@gmail.com>, Irwin King <king@cse.cuhk.edu.hk>.

*Proceedings of the 43rd International Conference on Machine Learning*, Seoul, South Korea. PMLR 306, 2026. Copyright 2026 by the author(s).

multimodal representation learning, such as MERU and related hyperbolic vision–language models (Desai et al., 2023; Mandica et al., 2024; Kim et al., 2024; Pal et al., 2025; Poppi et al., 2025), show that leveraging hyperbolic geometry leads to higher-quality multimodal representations. Specifically, hyperbolic space encodes semantic information along two complementary geometric dimensions: *(i) geodesic distances capture semantic similarity*, ensuring that concept proximity reflects relational closeness; *(ii) entailment cones encode semantic hierarchy*, organizing concepts according to their level of abstraction, with generic concepts typically located more centrally and more specific concepts extending outward along entailment cones (Ganea et al., 2018a; Desai et al., 2023; Pal et al., 2025).

Despite these advantages, existing hyperbolic multimodal models are almost exclusively developed under static learning assumptions, where the full data distribution is available upfront (Desai et al., 2023; Mandica et al., 2024; Kim et al., 2024; Pal et al., 2025; Poppi et al., 2025). In realistic multimodal systems, however, data distributions evolve over time as new concepts, domains, and tasks emerge (Kirkpatrick et al., 2017; Yu et al., 2024a; Ni et al., 2023; Liu et al., 2025a), leaving continual learning in hyperbolic representation spaces largely unexplored.

**Continual Learning as a Geometric Problem.** Because hyperbolic semantics are encoded by geometric quantities, continual parameter updates that violate their invariants distort semantic similarity and hierarchy, leading to catastrophic forgetting (Kirkpatrick et al., 2017; Lopez-Paz & Ranzato, 2017; Ni et al., 2023). This perspective highlights the fundamental challenge that need to be addressed: *what geometric conditions must be preserved to prevent forgetting in hyperbolic multimodal representations*.

**Contributions.** This work explores continual learning for hyperbolic multimodal representations and addresses the above fundamental questions. To the best of our knowledge, it provides the first geometric study of this setting. In particular, we make the following contributions:

- First, we establish a *theoretical foundation* for representation preservation in hyperbolic space, showing that it requires cross-modal invariance under a shared hyperbolic isometry (e.g., the same rotation), yielding necessary and sufficient geometric conditions.

- Second, guided by the above theoretical conditions and analysis, we derive a *principled continual learning framework* for hyperbolic multimodal representations, which preserves essential geometric structure while allowing effective adaptation to new tasks.

Across diverse continual multimodal benchmarks, HMCL consistently reduces catastrophic forgetting while adapting to new tasks.

## 2. Preliminary

### 2.1. Hyperbolic Geometry

Hyperbolic space is a non-Euclidean geometry with constant negative curvature. The Lorentz model of hyperbolic geometry (hyperboloid model) is widely used due to its numerical stability for optimization (Nickel & Kiela, 2018; Mishne et al., 2023). We provide a detailed introduction in Appendix B.2.

**Lorentz Model of Hyperbolic Space.** The $d$-dimensional hyperbolic space with curvature $-1/K$ ($K > 0$) can be realized as the upper sheet of a hyperboloid embedded in $(d+1)$-dimensional Minkowski space. Let $\mathbf{G} = \mathrm{diag}(-K, 1, \ldots, 1) \in \mathbb{R}^{(d+1)\times(d+1)}$ denote the Minkowski metric tensor. The hyperbolic manifold is defined as:

$$\mathbb{H}_K^d = \{\mathbf{x} \in \mathbb{R}^{d+1} \mid \mathbf{x}^\top \mathbf{G}\mathbf{x} = -K, \ x_0 > 0\}. \quad (1)$$

Each point $\mathbf{x} = [x_0; \mathbf{x}_{[1:d]}]^\top \in \mathbb{H}_K^d$ consists of a time-like coordinate $x_0$ and $d$ space-like coordinates $\mathbf{x}_{[1:d]}$, reflecting the Lorentzian signature of the metric.

**Lorentzian Inner Product and Distance.** The Lorentzian inner product is defined as:

$$\langle \mathbf{x}, \mathbf{y} \rangle_\mathcal{L} = \mathbf{x}^\top \mathbf{G}\mathbf{y} = -Kx_0y_0 + \sum_{i=1}^d x_iy_i, \quad (2)$$

which induces the geodesic distance:

$$\begin{aligned} d_\mathcal{L}^K(\mathbf{x}, \mathbf{y}) &= \sqrt{K} \operatorname{arcosh}\left(-\langle \mathbf{x}, \mathbf{y} \rangle_\mathcal{L}/K\right) \\ &= \sqrt{K} \operatorname{arcosh}\left(-\mathbf{x}^\top \mathbf{G}\mathbf{y}/K\right). \end{aligned} \quad (3)$$

**Lorentz Transformation Layer.** Let $\mathbf{z} \in \mathbb{R}^{d+1}$ be a Lorentz embedding, where $z_0 > 0$ denotes the time-like coordinate and $\mathbf{z}_s \in \mathbb{R}^d$ denotes the space-like coordinates. A Lorentz transformation layer (Chen et al., 2022) is defined as follows

$$f(\mathbf{W}; \mathbf{z}) = \begin{bmatrix} \sqrt{K + \|\mathbf{W}\mathbf{z}\|_2^2} \\ \mathbf{W}\mathbf{z} \end{bmatrix}, \quad (4)$$

where $\mathbf{W} \in \mathbb{R}^{d \times (d+1)}$, $K > 0$ is the curvature parameter of the Lorentz manifold. The transformation matrix is decomposed as $\mathbf{W} = \begin{bmatrix} \mathbf{w}_0 & \mathbf{W}_s \end{bmatrix}$, $\mathbf{w}_0 \in \mathbb{R}^{d \times 1}$, $\mathbf{W}_s \in \mathbb{R}^{d \times d}$, where $w_0$ and $W_s$ act on the time-like and space-like components of the input embedding, respectively.

### 2.2. Hyperbolic Multimodal Learning

Given a paired dataset $\mathcal{X}^{m,m'} = \{(\mathbf{x}_i^m, \mathbf{x}_i^{m'})_{i=1}^N\} \subset \mathcal{X}^m \times \mathcal{X}^{m'}$, where $N$ is the number of paired samples, $\mathcal{X}^m$ and $\mathcal{X}^{m'}$ represent the data spaces of modality $m$ and modality $m'$, respectively. Each pair $(\mathbf{x}_i^m, \mathbf{x}_i^{m'})$ consists of semantically corresponding samples from different modalities. The model produces embedding matrices

$\mathbf{Z}^m, \mathbf{Z}^{m'} \in \mathbb{R}^{N \times (d+1)}$, where each row $\mathbf{z}_i^m, \mathbf{z}_i^{m'} \in \mathbb{H}^d$ represents the hyperbolic embedding for the $i$-th sample pair.

**Contrastive Similarity.** The hyperbolic contrastive similarity matrix is computed as:

$$
\begin{aligned}
\mathbf{S}^{m \to m'} &= -d_{\mathcal{L}}^K(\mathbf{Z}^m, \mathbf{Z}^{m'}) \\
&= -\sqrt{K} \operatorname{arcosh}\left(-\mathbf{Z}^m \mathbf{G}(\mathbf{Z}^{m'})^\top / K\right),
\end{aligned} \quad (5)
$$

where $\mathbf{S}^{m \to m'} \in \mathbb{R}^{N \times N}$ denotes the similarity matrix, $\mathbf{G} = \operatorname{diag}(-K, 1, \dots, 1) \in \mathbb{R}^{(d+1) \times (d+1)}$ is the Minkowski metric tensor, and $\operatorname{arcosh}$ is applied element-wise. Each element $\mathbf{S}_{ij}^{m \to m'} = -d_{\mathcal{L}}^K(\mathbf{z}_i^m, \mathbf{z}_j^{m'})$ represents the negative hyperbolic distance between the $i$-th embedding from modality $m$ and the $j$-th embedding from modality $m'$.

The unidirectional contrastive loss is formulated as:

$$
\mathcal{L}_{m \to m'} = -\frac{1}{N} \sum_{i=1}^N \log \frac{\exp(\mathbf{S}_{ii}^{m \to m'} / \tau)}{\sum_{j=1}^N \exp(\mathbf{S}_{ij}^{m \to m'} / \tau)} \quad (6)
$$

where $\tau$ is the temperature parameter. The symmetric contrastive loss is defined as:

$$
\mathcal{L}_{\text{contrast}} = \frac{1}{2}\left(\mathcal{L}_{m \to m'} + \mathcal{L}_{m' \to m}\right) \quad (7)
$$

**Entailment cone.** For hierarchical relationships, each embedding $\mathbf{z}_i^m \in \mathbb{H}^d$ defines an entailment cone with aperture:

$$
\operatorname{aper}(\mathbf{z}_i^m) = \sin^{-1}\left(\frac{2\kappa}{\|(\mathbf{z}_i^m)_{[1:d]}\|_2}\right), \quad (8)
$$

where a constant $\kappa = 0.1$ is used for setting boundary conditions near the origin. The aperture naturally encodes semantic hierarchy: embeddings closer to the origin (i.e., smaller $\|(\mathbf{z}_i^m)_{[1:d]}\|_2$) correspond to higher-level, more general concepts and have larger apertures $\operatorname{aper}(\mathbf{z}_i^m)$, while embeddings farther from the origin represent more specific concepts with smaller apertures. The exterior angle between embeddings $\mathbf{z}_i^m$ and $\mathbf{z}_i^{m'}$ is:

$$
\operatorname{ext}(\mathbf{z}_i^m, \mathbf{z}_i^{m'}) = \cos^{-1}\left(\frac{(\mathbf{z}_i^{m'})_0 + (\mathbf{z}_i^m)_0 \langle \mathbf{z}_i^m, \mathbf{z}_i^{m'} \rangle_{\mathcal{L}}}{\|(\mathbf{z}_i^m)_{[1:d]}\|_2 \sqrt{\langle \mathbf{z}_i^m, \mathbf{z}_i^{m'} \rangle_{\mathcal{L}}^2 - 1}}\right). \quad (9)
$$

The entailment loss (Ganea et al., 2018b) enforces partial order constraints to ensure proper hierarchical relationships between cross-modal concepts (e.g., textual representations at higher semantic levels than their visual counterparts):

$$
\mathcal{L}_{\text{entail}}(\mathbf{z}_i^m, \mathbf{z}_i^{m'}) = \max(0, \operatorname{ext}(\mathbf{z}_i^m, \mathbf{z}_i^{m'}) - \operatorname{aper}(\mathbf{z}_i^m)). \quad (10)
$$

# 3. Problem Statement

## 3.1. Hyperbolic Multimodal Continual Framework

We consider multimodal continual learning under hyperbolic representations, in which multiple modalities are aligned into a common hyperbolic space and learned sequentially over time. The goal is to ensure that the learned embeddings preserve critical geometric relationships, both within and across modalities, while adapting to new tasks. We adopt the established continual learning framework for multi-modal contrastive learning (Liu et al., 2025b), where $M$ modalities are processed through pre-trained encoders $\mathcal{E}_m : \mathcal{X}_m \to \mathbb{R}^{d_m}$ followed by learnable transformation layers $T_m^{(t)} : \mathbb{R}^{d_m} \to \mathbb{R}^d$ for each modality $m \in \{1, 2, \dots, M\}$ at task $t$. The complete forward pass consists of:

**Features before continual learning:** The frozen encoder extracts features from task $t-1$ data:

$$
\mathbf{Z}_{t-1}^{m,*} = \mathcal{E}_m(X_{t-1}^m) \in \mathbb{R}^{N \times d_m} \quad (11)
$$

where $\mathcal{E}_m$ denotes the pre-trained encoder for modality $m$.

**Transformation layer:** To effectively accumulate knowledge through incremental updates, a learnable transformation layer adapts the features:

$$
\mathbf{Z}_{t-1}^{m,t} = T_m^{(t)}(\mathbf{Z}_{t-1}^{m,*}) = f(\mathbf{W}^{m,t}, \mathbf{Z}_{t-1}^{m,*}) \in \mathbb{R}^{N \times (d+1)} \quad (12)
$$

where $T_m^{(t)}$ denotes the transformation layer for modality $m$ at task $t$, $\mathbf{W}^{m,t}$ is a learnable weight matrix of the linear transformation stage $t$, and $f(\cdot, \cdot)$ is a function ensuring that the outputs still lie on the Lorentz manifold $\mathbb{H}^d$.

Similarly, for the current task $t$ data:

$$
\mathbf{Z}_t^{m,t} = T_m^{(t)}(\mathcal{E}_m(X_t^m)) \in \mathbb{R}^{N \times (d+1)} \quad (13)
$$

**Notation Justification.** In our notation system, $\mathbf{Z}_t^{m,s}$ represents the embedding of task $t$ data for modality $m$ processed at stage $s$. For example, $\mathbf{Z}_{t-1}^{m,*}$ – task $t-1$ data processed by frozen pre-trained encoder ($*$ indicates pre-trained weights); $\mathbf{Z}_{t-1}^{m,t}$ – task $t-1$ data processed by learnable transformation at stage $t$. We list the notation table in Appendix A.

## 3.2. Key Challenges in Hyperbolic Geometry

**Stability.** In hyperbolic representation learning, the preservation of historical knowledge fundamentally depends on maintaining the geometric structure of the embedding space. This requires maintaining three geometric properties of the learned representations[1]. $\forall m, m' \in \{1, 2, \dots, M\}, m \neq m'$:

---

[1]Given that $\operatorname{arccosh}(\cdot)$ is a strictly monotonic function in Equation (19), equality of similarity values $\mathbf{S}$ directly corresponds to equality of the underlying Lorentzian inner products in Definition 2. We thus simplify our notation accordingly.

**(P1) Intra-modal preservation**:
$$\mathbf{Z}_{t-1}^{m,t}\mathbf{G}(\mathbf{Z}_{t-1}^{m,t})^\top = \mathbf{Z}_{t-1}^{m,t-1}\mathbf{G}(\mathbf{Z}_{t-1}^{m,t-1})^\top;$$

**(P2) Inter-modal preservation**:
$$\mathbf{Z}_{t-1}^{m,t}\mathbf{G}(\mathbf{Z}_{t-1}^{m',t})^\top = \mathbf{Z}_{t-1}^{m,t-1}\mathbf{G}(\mathbf{Z}_{t-1}^{m',t-1})^\top;$$

**(P3) Hierarchical preservation**:
$$\|(\mathbf{z}_{t-1}^{m,t})_{[1:d]}\| = \|(\mathbf{z}_{t-1}^{m,t-1})_{[1:d]}\|.$$

*Intra-modal preservation* ensures that domain-specific knowledge within each modality remains intact, while *inter-modal preservation* maintains the learned cross-modal associations that enable effective multimodal understanding. *Hierarchical preservation* protects the conceptual hierarchies captured by the hyperbolic cone structure, which are crucial for representing complex semantic relationships between concepts (Detailed explanations are shown in Appendix D).

**Plasticity.** The model maintains the capability to learn from new datasets and adapt to evolving multimodal distributions. For the current task $t$, the model effectively acquires new knowledge by learning appropriate representations: $\{\mathbf{Z}_t^{m,t}\}_{m=1}^M \sim p_t^{(1,2,\ldots,M)}$, where $p_t^{(1,2,\ldots,M)}$ represents the joint distribution of the $t$-th task dataset across all $M$ modalities. This ensures the model can effectively capture task-specific patterns and cross-modal relationships while maintaining its representational capacity.

**Challenges.** The above formulations clarify *what* stability and plasticity demand in hyperbolic multimodal continual learning. The challenge that must be addressed to make these objectives achievable in practice is: *the stability conditions (P1)–(P3) are principled but cumbersome to enforce or analyze directly, motivating a simpler equivalent characterization (Theorem 1)* .

## 4. Principles of Geometric Updates

In this section, we derive geometric principles that determine the legality of parameter updates through preservation of geometric invariants in hyperbolic representations.

### 4.1. Geometric Characterization

This section provides a characterization of the conditions under which old-task representations are preserved across continual learning stages.

**Theorem 1** (Geometric Characterization of Preservation). *Assume a non-degeneracy condition for task $t-1$ representations at stage $t-1$. Then the preservation conditions (P1), (P2), and (P3) hold for task $t-1$ representations between stages $t-1$ and $t$ if and only if there exists a shared*

*transformation of the form*
$$\mathbf{R} = \begin{pmatrix} 1 & \mathbf{0}^\top \\ \mathbf{0} & \widetilde{\mathbf{R}} \end{pmatrix}, \qquad \widetilde{\mathbf{R}} \in \mathrm{SO}(d),$$

*such that, for all modalities $m \in \{1,\ldots,M\}$,*
$$\mathbf{Z}_{t-1}^{m,t} = \mathbf{Z}_{t-1}^{m,t-1}\mathbf{R}^\top.$$

*Proof sketch.* Conditions (P1) and (P2) preserve all required Lorentzian inner products for task $t-1$ between stages $t-1$ and $t$. By Theorem 2 (and Corollary 3) in Appendix E.1, this yields a unique *shared* Lorentz transformation $\mathbf{L} \in \mathrm{SO}^+(1,d)$ acting identically across modalities. Condition (P3) enforces spatial-norm invariance; Together, these give Theorem 1; the detailed proof is shown in Appendix E.1.

**Role of non-degeneracy.** The non-degeneracy condition is used to obtain the sharpest identifiable characterization of the preservation map. When old-task embeddings span the relevant Lorentzian subspace, the preserved Gram structure determines a shared isometry uniquely, making the preservation requirement maximally constrained. If some representation directions collapse or become redundant, the same preservation conditions still apply, but the compatible isometry may no longer be unique because the old-task geometry is less informative along those collapsed directions. Thus, non-degeneracy should be interpreted as an identifiability condition for the strongest theoretical statement, rather than as a restriction required by the HMCL algorithm.

### 4.2. Hierarchy Preservation at First Order

In hyperbolic space, the time-like coordinate is functionally constrained by the spatial components through the geometry of the Lorentz manifold. Leveraging this structural dependency, this section characterizes the first-order conditions under which hierarchical structure is preserved, forming the theoretical basis for the update rules derivation.

**Corollary 1** (Updates under hierarchical preservation). *Consider an update $\Delta\mathbf{z} = \mathbf{z}_{t-1}^{m,t} - \mathbf{z}_{t-1}^{m,t-1}$ applied to the old-task embedding $\mathbf{z}_{t-1}^{m,t-1}$. If the time-like coordinate is required to be first-order invariant, i.e. (P3), then the induced spatial update must satisfy*

$$\mathbf{z}_{[1:d]}^\top \Delta\mathbf{z}_{[1:d]} = 0. \tag{14}$$

**Key takeaway.** Corollary 1 indicates that preserving the hierarchy of old-task embeddings enforces that the spatial update is orthogonal to the current embedding's spatial direction. By Equation (30), admissible updates are restricted within the tangent space so as to suppress first-order drift along the time-like dimension.

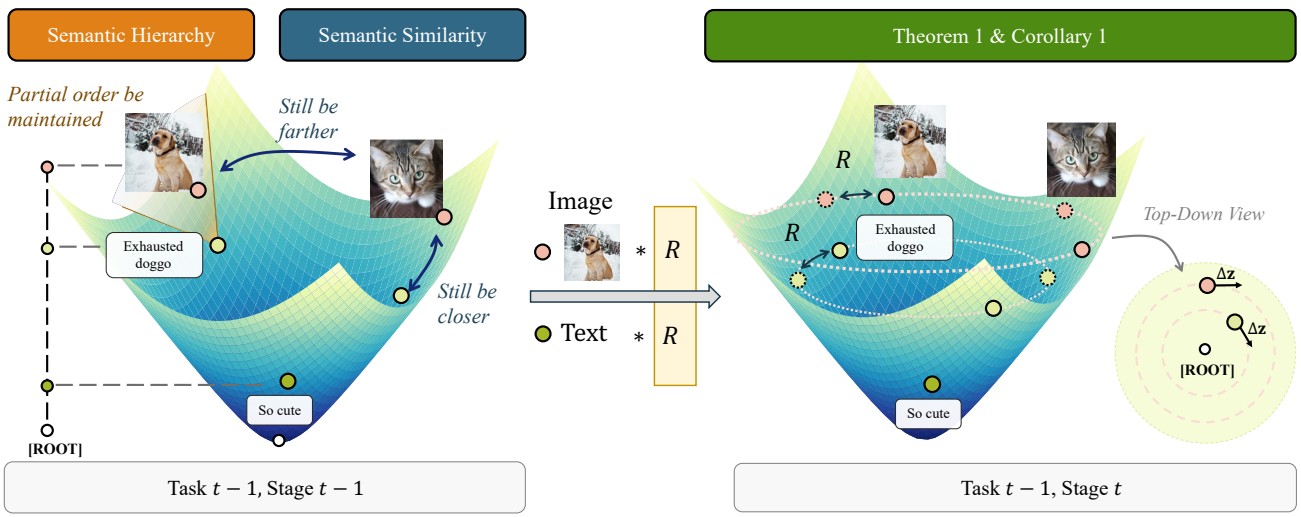

*Figure 2.* **Geometric interpretation of hyperbolic multimodal continual learning.** Multimodal representations are embedded in hyperbolic space, where distance encodes semantic similarity and distance to the origin encodes semantic hierarchy. Preserving old-task knowledge across tasks requires maintaining these geometric relations under a shared hyperbolic isometry. The top-down view illustrates that first-order hierarchy preservation restricts updates to lie within the tangent space.

**Geometric interpretation.** Figure 2 illustrates hyperbolic multimodal continual learning from a geometric perspective. In hyperbolic space, distances encode semantic similarity, while distance to the origin captures semantic hierarchy. As established in Theorem 1, preserving old-task knowledge across tasks requires updates to act as a shared hyperbolic isometry, maintaining both properties.

The top-down view highlights the consequence of Corollary 1: at first order, hierarchy preservation restricts updates to changes that keep representations' distance to the origin unchanged. Conventional continual learning methods lack such constraints, leading to violations of the stability conditions and degradation on previously learned tasks.

## 5. Hyperbolic Multimodal Continual Learning

Based on Theorem 1 and Corollary 1, we further derive a principled methodology for identifying admissible parameter updates that satisfy the geometric characterization established in this section.

### 5.1. Admissible Parameter Updates

**Proposition 1** (Admissible Parameter Updates). *Consider the Lorentz transformation layer in Section 2.1, with parameters $\mathbf{W}^{m,t} = [\mathbf{w}_0^{m,t} \quad \mathbf{W}_s^{m,t}]$. Assume Theorem 1 holds at stage $t-1$. For an infinitesimal update $\mathbf{W}^{m,t} = \mathbf{W}^{m,t-1} - \eta \Delta \mathbf{W}^m$ with $\eta \to 0^+$, $\Delta \mathbf{W}^m$ is admissible if and only if there exists $\mathbf{\Omega}^\top = -\mathbf{\Omega}$ such that*

$$\mathbf{Z}_{t-1}^{m,*}[1:d]\left(\Delta\mathbf{W}_s^m\right)^\top = \mathbf{Z}_{t-1}^{m,t-1}[1:d]\mathbf{\Omega}^\top. \quad (15)$$

*Moreover, admissibility requires $\mathbf{Z}_{t-1,0}^{m,*}\left(\Delta\mathbf{w}_0^m\right) = \mathbf{0}$, which implies $\Delta\mathbf{w}_0^m = \mathbf{0}$ under a mild non-degeneracy condition.*

**Corollary 2** (Block-wise Canonical Update). *Under the admissibility condition of Proposition 1, the canonical admissible update is characterized by* [2]

$$\mathbf{Z}_{t-1,s}^{m,*}\left(\Delta\mathbf{W}_s^m\right)^\top = \mathbf{0}, \qquad \Delta\mathbf{w}_0^m = \mathbf{0}, \quad (16)$$

*and is given in closed form by*

$$\Delta\mathbf{W}_s^m = \delta_s^m\left(\mathbf{I} - \mathbf{P}_{t-1}\right), \quad (17)$$

*where $\mathbf{P}_{t-1} = (\mathbf{Z}_{t-1,s}^{m,*})^\top\left(\mathbf{Z}_{t-1,s}^{m,*}(\mathbf{Z}_{t-1,s}^{m,*})^\top\right)^\dagger\mathbf{Z}_{t-1,s}^{m,*}$.*

The detailed proofs of Proposition 1 and Corollary 2 are provided in Appendix F.1 and Appendix F.2, respectively.

This section establishes a principled link between geometric preservation and parameter updates in hyperbolic multimodal continual learning.

> **Key takeaway.** We show that preserving intra-modal, inter-modal, and hierarchical structure in turn induces a strict first-order constraint on admissible parameter updates (Proposition 1). Under this constraint, the optimal update is given by an orthogonal projection, providing a closed-form and geometry-consistent parameter update rule (Corollary 2).

---

[2]To simplify notation, $\mathbf{Z}_{t-1}^{m,*}$ represents $\mathbf{Z}_{t-1}^{m,*}[1:d]$.

---

**Algorithm 1:** HMCL with Canonical Updates

**Input:** Old-task spatial representations
$\mathbf{Z}_{t-1,s}^{m,*} \in \mathbb{H}_K^d$;
Current parameters
$\mathbf{W}^{m,t-1} = [\mathbf{w}_0^{m,t-1}, \mathbf{W}_s^{m,t-1}]$;
Unconstrained update direction $\boldsymbol{\delta}_s^m$;
Step size $\eta$.

**Output:** Updated parameters $\mathbf{W}^{m,t}$.
Compute a low-rank orthonormal basis $\mathbf{V}$ of $\mathbf{Z}_{t-1,s}^{m,*}$
  via SVD (or PCA) (Lopez-Paz & Ranzato, 2017);
Project the unconstrained update:

$$\Delta \mathbf{W}_s^m = \boldsymbol{\delta}_s^m (\mathbf{I} - \mathbf{V}\mathbf{V}^\top);$$

Set the time-like update to zero $\Delta \mathbf{w}_0^m = \mathbf{0}$;
Update parameters:

$$\mathbf{W}_s^{m,t} = \mathbf{W}_s^{m,t-1} - \eta \, \Delta \mathbf{W}_s^m, \qquad \mathbf{w}_0^{m,t} = \mathbf{w}_0^{m,t-1}.$$

**return** $\mathbf{W}^{m,t}$.

---

### 5.2. Training with Canonical Updates (HMCL)

We now describe how the canonical update in Corollary 2 is incorporated into training. At each stage, the update restricts parameter changes to directions orthogonal to the subspace induced by old-task representations, while remaining compatible with standard gradient-based optimization. The procedure is summarized in Algorithm 1, which applies the canonical projection to the unconstrained update and enforces stability of hyperbolic old-task representations.

**Remark 1.** *The canonical update introduces no additional learnable parameters. The projection basis $\mathbf{V}$ depends only on old-task representations and can be computed once per stage. In implementation, we maintain a fixed-size $d \times d$ covariance summary of the retained old-task spatial subspace, rather than storing separate bases or historical representations for every previous task; hence the memory overhead is constant with respect to the number of tasks. As a result, the update incurs negligible overhead and can be seamlessly integrated into existing training pipelines.*

We discuss finite-step AdamW optimization and its residual approximation error in Appendix F.3.

## 6. Experiments

The experiments are designed to systematically evaluate continual learning for hyperbolic multimodal representations. We organize the evaluation around the following research questions.

- **RQ1:** Does HMCL improve performance and reduce forgetting across backbones? (Section 6.2)

- **RQ2:** Does HMCL reduce radial, angular, and cross-modal old-task drift? (Section 6.3)

- **RQ3:** Does HMCL preserve semantic hierarchies after continual training? (Section 6.4)

Supplementary ablations, robustness checks, and additional traversals are referenced below and reported in Appendix G.2, Appendix G.3, and Appendix G.4, respectively.

### 6.1. Implementation Details

**Tasks & datasets.** We evaluate under a continual multimodal learning setting with both classification and retrieval tasks. The task stream consists of up to 15 datasets, including image–text classification benchmarks (e.g., CIFAR-10/100 (Krizhevsky, 2009), Caltech-101 (Fei-Fei et al., 2004), Food-101 (Bossard et al., 2014)) and cross-modal retrieval benchmarks (COCO (Lin et al., 2014) and Flickr30k (Young et al., 2014)). Tasks are presented sequentially without revisiting previous data, using a fixed task order across all runs. Shuffled-stream results in Table 6 show that HMCL retains consistent gains under alternative task orders. Dataset statistics are provided in Appendix G.1.

**Implementation details.** We adopt pretrained multimodal encoders as frozen feature extractors and optimize lightweight transformation layers throughout continual training. All representations are modeled in the Lorentz (hyperboloid) space with fixed curvature $K = 0.1$. The transformation module consists of a single linear layer without activation or dropout.

Models are optimized using AdamW optimization (Loshchilov & Hutter, 2019). For classification tasks, we train for 10 epochs with a learning rate $5 \times 10^{-4}$ and batch size 1024. For retrieval tasks, we train for 15 epochs with a learning rate of $5 \times 10^{-5}$ and a batch size of 1024. The entailment loss weight is set to 0.2 for all tasks, and weight decay is set to 0. All experiments are implemented in PyTorch (Paszke et al., 2019) and conducted on a single NVIDIA RTX A6000 GPU. Table 5 further tests AdamW with nonzero weight decay and confirms that HMCL remains effective under this optimizer mismatch.

**Evaluation metrics.** For retrieval tasks, we report Recall@$k$ with $k = 5$ for both image-to-text and text-to-image directions. For classification tasks, we report average accuracy. To measure stability, we report backward transfer (BWT). All results are obtained by averaging over five independent runs with different random seeds.

**Baselines.** To the best of our knowledge, there is currently no prior work that specifically studies continual learning for

*Table 1.* Average performance (mean ± std.) on classification and retrieval tasks. Bold indicates statistical significance ($p < 0.05$).

| Backbone | Method | Classification | | Retrieval | | Overall | |
|---|---|---|---|---|---|---|---|
| | | Acc ↑ | BWT$_A$ ↑ | R@5 ↑ | BWT$_{R5}$ ↑ | Performance ↑ | BWT ↑ |
| MERU-L | Vanilla | $40.59 \pm 0.21$ | $-6.46 \pm 0.23$ | $31.97 \pm 0.31$ | $-6.46 \pm 0.23$ | $38.22 \pm 0.18$ | $-3.49 \pm 0.24$ |
| | EWC | $40.52 \pm 0.15$ | $-6.43 \pm 0.15$ | $32.96 \pm 0.08$ | $-3.53 \pm 0.25$ | $38.22 \pm 0.13$ | $-6.04 \pm 0.13$ |
| | GEM | $40.55 \pm 0.16$ | $-6.40 \pm 0.16$ | $33.00 \pm 0.07$ | $-3.50 \pm 0.24$ | $38.25 \pm 0.14$ | $-6.02 \pm 0.13$ |
| | C-FLAT | $39.28 \pm 0.30$ | $-6.23 \pm 0.27$ | $31.42 \pm 0.17$ | $-2.56 \pm 0.30$ | $37.16 \pm 0.27$ | $-5.75 \pm 0.23$ |
| | HMCL (ours) | $\mathbf{43.45 \pm 0.09}$ | $\mathbf{-1.65 \pm 0.10}$ | $\mathbf{34.89 \pm 0.37}$ | $\mathbf{-1.65 \pm 0.10}$ | $\mathbf{41.03 \pm 0.16}$ | $\mathbf{-2.20 \pm 0.34}$ |
| | Δ vs. Vanilla | +7.0% | +74.5% | +9.1% | +74.5% | +7.4% | +37.0% |
| MERU-B | Vanilla | $44.34 \pm 0.07$ | $-7.26 \pm 0.14$ | $33.66 \pm 0.19$ | $-5.43 \pm 0.19$ | $41.82 \pm 0.07$ | $-5.43 \pm 0.19$ |
| | EWC | $44.48 \pm 0.28$ | $-6.63 \pm 0.28$ | $31.87 \pm 0.45$ | $-4.36 \pm 0.39$ | $41.94 \pm 0.22$ | $-6.34 \pm 0.25$ |
| | GEM | $44.91 \pm 0.76$ | $-7.00 \pm 0.72$ | $29.06 \pm 3.00$ | $-7.15 \pm 3.11$ | $42.09 \pm 0.55$ | $-6.81 \pm 0.73$ |
| | C-FLAT | $44.88 \pm 0.54$ | $-7.58 \pm 0.61$ | $28.20 \pm 3.16$ | $-7.91 \pm 3.55$ | $42.04 \pm 0.23$ | $-7.34 \pm 0.70$ |
| | HMCL (ours) | $\mathbf{46.25 \pm 0.07}$ | $\mathbf{-3.83 \pm 0.08}$ | $\mathbf{36.29 \pm 0.05}$ | $\mathbf{-2.78 \pm 0.10}$ | $\mathbf{43.69 \pm 0.07}$ | $\mathbf{-2.77 \pm 0.30}$ |
| | Δ vs. Vanilla | +4.3% | +47.2% | +7.8% | +48.8% | +4.5% | +49.0% |
| HyCoCLIP-B | Vanilla | $40.70 \pm 0.20$ | $-5.46 \pm 0.14$ | $45.34 \pm 0.35$ | $-5.46 \pm 0.14$ | $42.33 \pm 0.20$ | $-5.42 \pm 0.63$ |
| | EWC | $40.54 \pm 0.24$ | $-5.51 \pm 0.14$ | $45.62 \pm 0.35$ | $-4.52 \pm 0.31$ | $42.25 \pm 0.22$ | $-5.53 \pm 0.12$ |
| | GEM | $40.58 \pm 0.27$ | $-5.48 \pm 0.16$ | $45.67 \pm 0.41$ | $-4.46 \pm 0.36$ | $42.29 \pm 0.24$ | $-5.51 \pm 0.14$ |
| | C-FLAT | $40.67 \pm 0.18$ | $-5.49 \pm 0.13$ | $45.31 \pm 0.28$ | $-4.52 \pm 0.24$ | $42.32 \pm 0.17$ | $-5.50 \pm 0.12$ |
| | HMCL (ours) | $\mathbf{42.33 \pm 0.10}$ | $\mathbf{-2.02 \pm 0.14}$ | $\mathbf{49.43 \pm 0.05}$ | $\mathbf{-1.73 \pm 0.10}$ | $\mathbf{44.13 \pm 0.12}$ | $\mathbf{-3.03 \pm 0.35}$ |
| | Δ vs. Vanilla | +4.0% | +63.0% | +9.0% | +68.3% | +4.3% | +44.1% |

hyperbolic multimodal representations. We therefore adapt representative continual learning strategies originally developed in Euclidean space for comparison. (1) *Vanilla.* The Vanilla baseline sequentially fine-tunes the hyperbolic multimodal model on each task using the standard contrastive objective, without any explicit mechanism to mitigate forgetting. Naive continual training in vision–language models is known to suffer from severe cross-modal misalignment and spatial disorder (Ni et al., 2023). (2) *EWC and GEM.* We include representative regularization- and constraint-based methods, namely Elastic Weight Consolidation (EWC) and Gradient Episodic Memory (GEM), which are commonly adopted baselines in multimodal continual learning (Ni et al., 2023). Both methods are applied directly to the hyperbolic multimodal training objective. (3) *C-FLAT.* C-FLAT introduces contrastive regularization to encourage consistency between current and historical representations across tasks in continual vision–language learning (Bian et al., 2024). We adopt the original formulation and apply it directly in the hyperbolic setting. For all baselines, we replace the original Euclidean training objective with the hyperbolic contrastive loss to better match the underlying geometry. All other continual learning strategies remain unchanged, and we adopt their original formulations and apply them directly in the hyperbolic setting. This protocol compares the continual-learning mechanisms under the same hyperbolic objective and backbone. However, these baselines

remain geometry-agnostic in their update rules: they do not explicitly preserve shared cross-modal isometries or hierarchy-related constraints, which is precisely the gap targeted by HMCL.

### 6.2. Main Results (RQ1)

**Experimental Results. HMCL consistently improves overall performance across backbones.** Table 1 reports average performance on continual multimodal classification and retrieval across three hyperbolic backbones (MERU-L, MERU-B, and HyCoCLIP-B). Across all settings, HMCL achieves the highest classification accuracy, retrieval performance (R@5), and overall score. For example, on MERU-L, HMCL improves the Overall metric from 38.22 to 41.03, corresponding to a relative gain of +7.4% over the Vanilla baseline. On HyCoCLIP-B, HMCL increases retrieval R@5 by +9.0%. These consistent gains indicate that the effectiveness of HMCL generalizes across model architectures and capacities.

**HMCL substantially mitigates catastrophic forgetting.** In terms of backward transfer (BWT), HMCL consistently yields significantly less negative values than competing approaches. On MERU-L, classification BWT improves from $-6.46$ to $-1.65$, corresponding to a 74.5% relative reduction in forgetting. Similar improvements are observed for retrieval BWT and across all three backbones. In contrast,

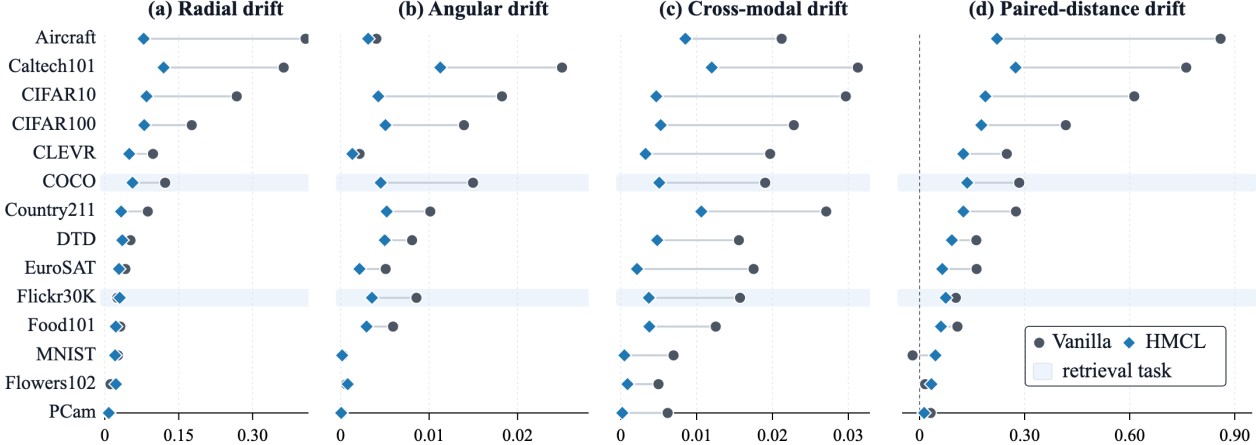

*Figure 3.* Dataset-wise old-task representation drift from the task-end checkpoint to the final checkpoint. Gray circles denote Vanilla and blue diamonds denote HMCL; shaded rows indicate retrieval tasks. We measure four complementary changes: radial drift for hierarchy-related radius changes, angular drift for within-modality directional distortion, cross-modal drift for image–text relational structure, and paired-distance drift for matched image–text geodesic distances. Lower values indicate better preservation.

EWC and GEM provide only marginal relief, while C-FLAT alleviates forgetting in some cases but often at the expense of overall performance.

**Conventional continual learning methods struggle in hyperbolic multimodal settings.** Regularization-based methods such as EWC and constraint-based approaches like GEM show limited effectiveness and, in some cases, high variance. C-FLAT partially improves retrieval stability but fails to consistently preserve classification and overall performance. These results suggest that methods developed under Euclidean assumptions are insufficient when continual updates must preserve hyperbolic invariants governing both cross-modal relations and hierarchy.

**Geometry-aware constraints are key to stable multimodal continual learning.** The consistent advantages of HMCL align with the theoretical analysis: by restricting continual updates to shared hyperbolic isometries, HMCL preserves both cross-modal similarity and hierarchical structure. This geometric consistency enables simultaneous improvements in classification and retrieval, rather than trading off performance across tasks. Figure 5 complements this result by showing that removing the block-wise constraint weakens stability across backbones.

### 6.3. Quantitative Drift Analysis (RQ2)

**Protocol.** To directly connect forgetting with geometric representation change, we measure old-task drift after subsequent continual updates. The four metrics below are organized by the preservation conditions in Section 3.

- **Radial drift** measures changes in hyperbolic radius, i.e., the spatial-norm/time-like signal tied to semantic hierarchy. It diagnoses violations of **hierarchical preservation** in **(P3)**.
- **Angular drift** removes magnitude effects and measures within-modality directional distortion. It reflects whether **intra-modal neighborhoods** remain stable as required by **(P1)**.
- **Cross-modal drift** measures changes in the image–text relational structure. It corresponds to **inter-modal preservation** in **(P2)**.
- **Paired-distance drift** tracks geodesic-distance changes of matched image–text pairs. It directly tests **cross-modal semantic alignment** under **(P2)**, while remaining sensitive to hierarchy-related radial changes under **(P3)**.

**Results.** As shown in Figure 3, HMCL reduces all four drift types for most datasets, including both classification and retrieval tasks. Averaged over old tasks, the reductions are 61.0%, 57.8%, 73.8%, and 59.5%, respectively, with aggregate statistics and significance in Table 7. **Takeaway:** forgetting involves both hierarchy-related distortion and relation-level drift.

### 6.4. Case Study (RQ3)

**Data Preparation.** To explore the efficacy of our method in preserving hierarchical relationships within hyperbolic space during continual learning, we conduct a detailed case study using image traversals on the Flickr30K dataset. We construct a hierarchical text set by extracting nouns and

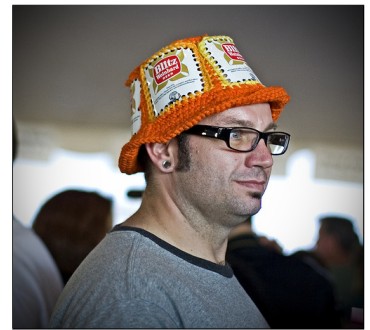 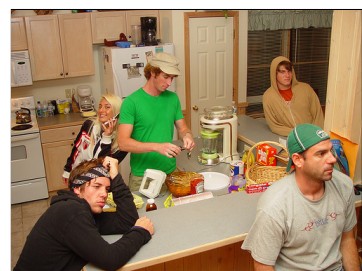 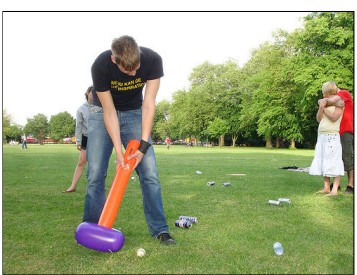

| HMCL(ours) | Vanilla | HMCL(ours) | Vanilla | HMCL(ours) | Vanilla |
|:---:|:---:|:---:|:---:|:---:|:---:|
| *hat* | *volleyball* | *relaxation* | *relaxation* | *defeat* | *workout* |
| *fashion* | *interest* | *vacation* | *aloof* | *workout* | *pilates* |
| *style* | *fashion* | *peace* | *sunken* | *peace* | *race* |
| **[ROOT]** | **[ROOT]** | **[ROOT]** | **[ROOT]** | **[ROOT]** | **[ROOT]** |

*Figure 4.* **Case study: image traversals and hierarchical concept retrieval.** For three Flickr30K images, we traverse from each image embedding to the manifold origin [ROOT] along the geodesic and report the nearest text concept at each step (specific → generic). HMCL (ours) yields more coherent and semantically aligned concept hierarchies (e.g., *hat → fashion → style*); Vanilla often drifts to less relevant or inconsistent concepts after continual training.

adjectives from the original captions to represent varying levels of semantic abstraction, following the dataset details in Appendix G.1.

**Image Traversals.** We utilize the image traversal technique to qualitatively evaluate the structural integrity of the learned representation space. This method involves performing a shortest-path traversal along the geodesic connecting a specific image embedding to the manifold's origin, denoted as [ROOT], which represents the most generic semantic concept. The traversal is executed by interpolating 20 equally spaced steps in the tangent space at the origin. These interpolated vectors are subsequently projected back onto the Lorentz hyperboloid using the exponential map. At each discrete step, we perform nearest-neighbor retrieval from the candidate set of text embeddings.

**Results and Analysis.** In experiments focusing solely on nouns and adjectives, Figure 4 illustrates the comparison between HMCL and the Vanilla baseline model after continuous learning. For example, in the left image, the most salient concrete concept is a hat: HMCL follows a coherent path from "hat" to broader categories such as "fashion" and "style" as it approaches [ROOT]. Vanilla instead jumps to the unrelated concept "volleyball" before drifting through "interest" and "fashion", indicating a disrupted local-to-generic hierarchy. The traversal paths thus provide a qualitative counterpart to the drift analysis: HMCL keeps image-specific concepts connected to stable generic anchors as embeddings move toward [ROOT]. This suggests that HMCL preserves not only nearest-neighbor retrieval quality, but also the abstraction ordering encoded by the hyperbolic radial geometry, consistent with Appendix G.4.

## 7. Conclusion

This work presents the first systematic study of continual learning for hyperbolic multimodal representations. We establish a theoretical foundation showing that preserving previously learned knowledge requires cross-modal invariance under a shared hyperbolic isometry, and we clarify that forgetting in this setting involves both relation-level drift and hierarchy-related distortion, motivating updates that preserve both cross-modal relational structure and hierarchical geometry. We derive HMCL, a principled framework that restricts parameter updates to geometry-preserving directions. Experiments across multiple hyperbolic backbones demonstrate that HMCL consistently improves both performance and stability, substantially reducing catastrophic forgetting compared to existing continual learning methods.

**Limitation and Future directions.** This study follows a hyperbolic geometry-preserving route: HMCL constrains admissible updates to preserve relational and hierarchical structure in Lorentz space, rather than using replay. Combining this route with replay-style mechanisms, such as exemplar/generative replay or hyperbolic memory selection, and extending it to full-backbone adaptation remain promising directions.

**Acknowledgments.** Jiahong Liu and Irwin King acknowledge partial support from RGC of Hong Kong SAR, China (CUHK 2300246, RGC C1043-24G; CUHK 14203425, RGC GRF 2151317). Menglin Yang acknowledges partial support from the Guangdong Provincial Natural Science Foundation General Program (Grant No. 2026A1515012118).

## Impact Statement

This paper presents work whose goal is to advance the field of Machine Learning. There are many potential societal consequences of our work, none of which we feel must be specifically highlighted here. The main limitations and future directions of the technical contribution are discussed in the conclusion.

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

# Appendix

## Contents

## A. Notation Table

For reference, we provide a comprehensive summary of the notation used throughout this paper.

*Table 2.* Summary of notation used throughout the paper.

| Symbol | Description |
|---|---|
| **Hyperbolic geometry and the Lorentz model** | |
| $d$ | Dimension of hyperbolic space |
| $K$ | Curvature parameter ($K > 0$); the sectional curvature is $-1/K$ |
| $\mathbb{H}_K^d$ | $d$-dimensional hyperbolic space with curvature $-1/K$ |
| $\mathcal{L}_K^d$ | Lorentz manifold ($\mathbb{H}_K^d, \mathbf{G}$) with metric structure |
| $\mathbf{G}$ | Minkowski metric tensor, $\mathbf{G} = \mathrm{diag}(-K, 1, \ldots, 1) \in \mathbb{R}^{(d+1) \times (d+1)}$ |
| $\langle \cdot, \cdot \rangle_{\mathcal{L}}$ | Lorentzian inner product: $\langle \mathbf{x}, \mathbf{y} \rangle_{\mathcal{L}} = \mathbf{x}^\top \mathbf{G} \mathbf{y} = -K x_0 y_0 + \sum_{i=1}^d x_i y_i$ |
| $d_{\mathcal{L}}^K(\mathbf{x}, \mathbf{y})$ | Geodesic distance: $d_{\mathcal{L}}^K(\mathbf{x}, \mathbf{y}) = \sqrt{K} \, \mathrm{arcosh}\left(-\langle \mathbf{x}, \mathbf{y} \rangle_{\mathcal{L}} / K\right)$ |
| $\mathbf{o}$ | Origin of the Lorentz model: $\mathbf{o} = (\sqrt{K}, 0, \ldots, 0)^\top \in \mathbb{H}_K^d$ |
| $x_0$ | Time-like coordinate (first component of $\mathbf{x}$) |
| $\mathbf{x}_{[1:d]}$ | Space-like coordinates: $(x_1, \ldots, x_d)^\top \in \mathbb{R}^d$ |
| $\mathrm{SO}^+(1, d)$ | Proper orthochronous Lorentz group |
| $\mathrm{SO}(d)$ | Special orthogonal group (rotation group) on $\mathbb{R}^d$ |
| **Multimodal learning** | |
| $M$ | Number of modalities |
| $m, m'$ | Modality indices, $m, m' \in \{1, 2, \ldots, M\}$ |
| $N$ | Number of samples (batch size) |
| $\mathcal{X}^m$ | Data space of modality $m$ |
| $\mathcal{X}^{m,m'}$ | Paired dataset: $\{(\mathbf{x}_i^m, \mathbf{x}_i^{m'})\}_{i=1}^N$ |
| $\mathbf{x}_i^m$ | The $i$-th input sample from modality $m$ |
| $\mathbf{z}_i^m$ | Hyperbolic embedding of the $i$-th sample from modality $m$, where $\mathbf{z}_i^m \in \mathbb{H}_K^d$ |
| **Continual learning framework** | |
| $t$ | Task/stage index |
| $T_m^{(t)}$ | Transformation layer for modality $m$ at task $t$ |
| $\mathbf{W}^{m,t}$ | Learnable weight matrix for modality $m$ at stage $t$ |
| $X_t^m$ | Input data for task $t$, modality $m$ |
| $\mathbf{Z}_{t-1}^{m,*}$ | Task $t-1$ embeddings from the frozen pre-trained encoder ($*$ indicates pre-trained weights) |
| $\mathbf{Z}_{t-1}^{m,t-1}$ | Task $t-1$ embeddings processed by the transformation layer at stage $t-1$ |
| $\mathbf{Z}_{t-1}^{m,t}$ | Task $t-1$ embeddings processed by the transformation layer at stage $t$ |
| $\mathbf{Z}_t^{m,t}$ | Task $t$ embeddings processed by the transformation layer at stage $t$ |
| $f(\mathbf{W}, \cdot)$ | Transformation function ensuring outputs lie on the Lorentz manifold $\mathbb{H}_K^d$ |
| **Similarity measures and loss functions** | |
| $\mathbf{S}^{m \to m'}$ | Similarity matrix from modality $m$ to $m'$: $\mathbf{S}^{m \to m'} \in \mathbb{R}^{N \times N}$ |
| $\mathbf{S}_{ij}^{m \to m'}$ | Pairwise similarity (negative distance): $-d_{\mathcal{L}}^K(\mathbf{z}_i^m, \mathbf{z}_j^{m'})$ |
| $\tau$ | Temperature parameter for contrastive loss |
| $\mathcal{L}_{m \to m'}$ | Unidirectional contrastive loss from modality $m$ to $m'$ |
| $\mathcal{L}_{\mathrm{contrast}}$ | Symmetric contrastive loss: $\frac{1}{2}(\mathcal{L}_{m \to m'} + \mathcal{L}_{m' \to m})$ |
| $\mathrm{aper}(\mathbf{z})$ | Entailment cone aperture (half-angle) at embedding $\mathbf{z}$ |
| $\mathrm{ext}(\mathbf{z}, \mathbf{z}')$ | Exterior angle between embeddings $\mathbf{z}$ and $\mathbf{z}'$ |
| $\kappa$ | Boundary constant for entailment cones (typically $\kappa = 0.1$) |
| $\mathcal{L}_{\mathrm{entail}}$ | Entailment loss for enforcing hierarchical relationships |
| **Theoretical analysis and parameter updates** | |
| $\mathbf{L}$ | General Lorentz transformation, $\mathbf{L} \in \mathrm{SO}^+(1, d)$ |
| $\mathbf{R}$ | Spatial rotation matrix: $\mathbf{R} = \begin{pmatrix} 1 & \mathbf{0}^\top \\ \mathbf{0} & \widetilde{\mathbf{R}} \end{pmatrix}$ with $\widetilde{\mathbf{R}} \in \mathrm{SO}(d)$ |
| $\widetilde{\mathbf{R}}$ | Rotation matrix in $\mathrm{SO}(d)$ (spatial block of $\mathbf{R}$) |
| $\boldsymbol{\delta}^m$ | Unconstrained update direction (e.g., gradient) for modality $m$ |
| $\eta$ | Learning rate / stepsize |

# B. Preliminaries

## B.1. Overview of Hyperbolic Space

Hyperbolic space is a non-Euclidean geometry characterized by constant negative curvature. Unlike Euclidean space, which has zero curvature (flat), or spherical geometry, which has positive curvature, hyperbolic space curves "away from itself" at every point. This fundamental property gives hyperbolic space several distinctive characteristics:

**Exponential growth of volume:** In $n$-dimensional hyperbolic space $\mathbb{H}^n$ with curvature $-1$, the volume of a ball of radius $r$ grows as $V_{\mathbb{H}^n}(r) \propto \int_0^r \sinh^{n-1}(t)\, dt$, which behaves asymptotically as $e^{(n-1)r}$ for large $r$. In contrast, the volume of a Euclidean ball grows polynomially as $r^n$. For instance, in the hyperbolic plane ($n = 2$), the area of a disk is $A(r) = 2\pi(\cosh(r) - 1)$, compared to $\pi r^2$ in the Euclidean plane.

**Natural hierarchy representation:** The exponential growth property makes hyperbolic space particularly well-suited for embedding hierarchical structures such as trees, taxonomies, and knowledge graphs with low distortion (Nickel & Kiela, 2017; Sarkar, 2011).

## B.2. Lorentz Manifold: Formal Definitions

The Lorentz model realizes $d$-dimensional hyperbolic space as a hyperboloid embedded in $(d + 1)$-dimensional Minkowski space. We now provide the formal definitions.

**Definition 1** (Minkowski Space and Metric Tensor). *The $(d + 1)$-dimensional **Minkowski space** is the vector space $\mathbb{R}^{d+1}$ equipped with the indefinite metric tensor*

$$\mathbf{G} = \mathrm{diag}(-K, 1, \ldots, 1) \in \mathbb{R}^{(d+1)\times(d+1)},$$

*where $K > 0$ is a parameter related to the curvature of the hyperbolic space. The metric tensor $\mathbf{G}$ has signature $(-, +, \ldots, +)$, meaning it has one negative eigenvalue corresponding to the **time dimension** and $d$ positive eigenvalues corresponding to the **space dimensions**.*

**Definition 2** (Lorentzian Inner Product). *The **Lorentzian inner product** (also called the Minkowski inner product) for vectors $\mathbf{x}, \mathbf{y} \in \mathbb{R}^{d+1}$ is defined as:*

$$\langle \mathbf{x}, \mathbf{y} \rangle_{\mathcal{L}} = \mathbf{x}^\top \mathbf{G} \mathbf{y} = -K x_0 y_0 + \sum_{i=1}^{d} x_i y_i.$$

**Definition 3** (Lorentz Manifold). *The $d$-dimensional **Lorentz manifold** (or hyperboloid model of hyperbolic space) with curvature $-1/K$ ($K > 0$) is defined as:*

$$\mathbb{H}_K^d = \{\mathbf{x} \in \mathbb{R}^{d+1} \mid \langle \mathbf{x}, \mathbf{x} \rangle_{\mathcal{L}} = -K,\ x_0 > 0\} = \{\mathbf{x} \in \mathbb{R}^{d+1} \mid \mathbf{x}^\top \mathbf{G} \mathbf{x} = -K,\ x_0 > 0\}.$$

*We denote the Lorentz manifold with its metric structure as $\mathcal{L}_K^d = (\mathbb{H}_K^d, \mathbf{G})$.*

**Remark 2.** *The constraint $\mathbf{x}^\top \mathbf{G} \mathbf{x} = -K$ expands to:*

$$-K x_0^2 + x_1^2 + x_2^2 + \cdots + x_d^2 = -K,$$

*which can be rewritten as:*

$$x_0^2 - \frac{1}{K}(x_1^2 + x_2^2 + \cdots + x_d^2) = 1.$$

*This is the equation of a two-sheeted hyperboloid in $\mathbb{R}^{d+1}$. The condition $x_0 > 0$ selects the upper sheet, which forms a connected $d$-dimensional manifold. For any point $\mathbf{x} = (x_0, x_1, \ldots, x_d)^\top \in \mathbb{H}_K^d$, we have:*

$$x_0 = \sqrt{1 + \|\mathbf{x}_{[1:d]}\|^2/K} \geq 1,$$

*where $\mathbf{x}_{[1:d]} = (x_1, \ldots, x_d)^\top$ denotes the space dimensions.*

**Definition 4** (Induced Norm to the Origin (Desai et al., 2023)). *Given the origin of the Lorentz manifold $\mathbf{o} = (1, 0, \ldots, 0) \in \mathcal{L}_K^d$ and $\mathbf{x} \in \mathbb{H}^d$, the induced distance to the origin is*

$$d_{\mathcal{L}}(\mathbf{x}, \mathbf{o}) = \mathrm{arcosh}(x_0) = \mathrm{arcosh}\left(\sqrt{1 + \|\mathbf{x}_{[1:d]}\|^2}\right). \tag{18}$$

A fundamental concept in any geometric space is the notion of distance. In hyperbolic space, the geodesic distance (the length of the shortest path between two points) is given by the following formula.

**Definition 5** (Lorentzian Distance). *The **geodesic distance** between two points $\mathbf{x}, \mathbf{y} \in \mathcal{L}_K^d$ is:*

$$d_{\mathcal{L}}^K(\mathbf{x}, \mathbf{y}) = \sqrt{K} \operatorname{arcosh}\left(-\frac{\langle \mathbf{x}, \mathbf{y} \rangle_{\mathcal{L}}}{K}\right) = \sqrt{K} \operatorname{arcosh}\left(-\frac{\mathbf{x}^\top G \mathbf{y}}{K}\right). \tag{19}$$

**Remark 3.** *To verify that this formula is well-defined, observe that for any $\mathbf{x}, \mathbf{y} \in \mathbb{H}_K^d$:*

- *The Lorentzian inner product satisfies $\langle \mathbf{x}, \mathbf{y} \rangle_{\mathcal{L}} \leq -K$ (with equality if and only if $\mathbf{x} = \mathbf{y}$), which follows from the Cauchy-Schwarz inequality in Minkowski space.*

- *Therefore, $-\langle \mathbf{x}, \mathbf{y} \rangle_{\mathcal{L}}/K \geq 1$, ensuring that the argument of $\operatorname{arcosh}$ is in its valid domain $[1, \infty)$.*

- *When $\mathbf{x} = \mathbf{y}$, we have $-\langle \mathbf{x}, \mathbf{x} \rangle_{\mathcal{L}}/K = K/K = 1$, giving $d_{\mathcal{L}}^K(\mathbf{x}, \mathbf{x}) = \sqrt{K} \operatorname{arcosh}(1) = 0$.*

**Remark 4** (Comparison with Euclidean Distance). *The hyperbolic distance grows much more slowly than the Euclidean distance in the ambient space. For points far from the origin, the hyperbolic distance grows approximately logarithmically with the Euclidean distance. This property allows hyperbolic space to embed exponentially growing structures (like trees) with bounded distortion.*

The symmetries of the Lorentz model of hyperbolic geometry are described by the Lorentz group, which consists of all linear transformations that preserve the Lorentzian inner product.

**Definition 6** (Proper Orthochronous Lorentz Group). *The **proper orthochronous Lorentz group** is defined as:*

$$\mathrm{SO}^+(1, d) := \left\{ \mathbf{L} \in \mathbb{R}^{(d+1)\times(d+1)} : \mathbf{L}^\top G \mathbf{L} = G, \ \det(\mathbf{L}) = 1, \ L_{00} \geq 1 \right\},$$

*where $G = \operatorname{diag}(-K, 1, \dots, 1)$ is the Lorentz metric tensor.*

**Remark 5.** *The three conditions in the definition have the following interpretations:*

- *$\mathbf{L}^\top G \mathbf{L} = G$: The transformation preserves the Lorentzian inner product, i.e., $\langle \mathbf{L}\mathbf{x}, \mathbf{L}\mathbf{y} \rangle_{\mathcal{L}} = \langle \mathbf{x}, \mathbf{y} \rangle_{\mathcal{L}}$ for all $\mathbf{x}, \mathbf{y}$.*

- *$\det(\mathbf{L}) = 1$: The transformation is **proper** (orientation-preserving), excluding reflections.*

- *$L_{00} \geq 1$: The transformation is **orthochronous** (time-orientation preserving), ensuring that the upper sheet of the hyperboloid is mapped to itself.*

*The Lorentz group acts transitively on $\mathbb{H}_K^d$: for any two points $\mathbf{x}, \mathbf{y} \in \mathbb{H}_K^d$, there exists $\mathbf{L} \in \mathrm{SO}^+(1, d)$ such that $\mathbf{L}\mathbf{x} = \mathbf{y}$.*

### B.3. Lorentz Neural Networks

**Definition 7** (Lorentz Transformation Layer (Chen et al., 2022)). *Let $\mathbf{z} \in \mathbb{R}^{d+1}$ be a Lorentz embedding, where $\mathbf{z}_0 > 0$ denotes the time-like coordinate and $\mathbf{z}_s \in \mathbb{R}^d$ denotes the space-like coordinates. A Lorentz transformation layer is defined as*

$$f(\mathbf{W}; \mathbf{z}) = \begin{bmatrix} \sqrt{K + \|\mathbf{W}\mathbf{z}\|_2^2} \\ \mathbf{W}\mathbf{z} \end{bmatrix}, \qquad \mathbf{W} \in \mathbb{R}^{d \times (d+1)}, \tag{20}$$

*where $K > 0$ is the curvature parameter of the Lorentz manifold.*

**Remark 6.** *The time-like coordinate in Equation (20) is reconstructed deterministically from the spatial component to satisfy the Lorentz constraint.*

We further decompose the transformation matrix as

$$\mathbf{W} = \begin{bmatrix} \mathbf{w}_0 & \mathbf{W}_s \end{bmatrix}, \qquad \mathbf{w}_0 \in \mathbb{R}^{d \times 1}, \ \mathbf{W}_s \in \mathbb{R}^{d \times d}, \tag{21}$$

where $\mathbf{w}_0$ and $\mathbf{W}_s$ act on the time-like and space-like components of the input embedding, respectively.

# C. Related Work

## C.1. Hyperbolic Multimodal Learning

Hyperbolic geometry is a powerful framework for representation learning on hierarchical and scale-free data (Krioukov et al., 2010; Nickel & Kiela, 2017; 2018; Ganea et al., 2018b; Shimizu et al., 2020; Chami et al., 2019; Yang et al., 2025b; He et al., 2025). Recent work has broadened hyperbolic modeling beyond graph embeddings to recommendation, retrieval, fully hyperbolic transformers, foundation-model fine-tuning, and federated adaptation (Yang et al., 2022b;a; Qiu et al., 2024; Yang et al., 2024b; 2025a; Liu et al., 2024). In parallel, hyperbolic contrastive learning has been studied for graph embeddings and anomaly detection, including analyses of dimensional collapse in curved contrastive spaces (Liu et al., 2022; Zhang et al., 2025; Fu et al., 2024). Hyperbolic geometry has recently been introduced to multimodal representation learning to capture the inherent hierarchical structure between visual and linguistic concepts.

Desai et al. (Desai et al., 2023) proposed MERU, the first large-scale hyperbolic vision-language model that projects CLIP-style embeddings onto the Lorentz hyperboloid. Subsequent works have extended hyperbolic multimodal learning along several directions. Pal et al. (Pal et al., 2025) introduced compositional entailment learning to capture intra-modal hierarchies, modeling how complex scenes compose from simpler visual primitives and how compound phrases relate to atomic concepts. Yang et al. (Yang et al., 2024a) further investigated hyperbolic cones for implicit hierarchical learning without requiring explicit parent-child supervision. For scaling to larger architectures, Mandica et al. (Mandica et al., 2024) investigated hyperbolic embeddings in billion-parameter models like BLIP-2. Kim et al. (Kim et al., 2024) leveraged hyperbolic entailment cones for data filtering, using image specificity metrics derived from cone apertures to remove underspecified samples from pretraining datasets. More recently, Poppi et al. (Poppi et al., 2025) proposed HySAC, which exploits hyperbolic geometry to model safety hierarchies in vision-language models. Beyond image-text pretraining, hyperbolic multimodal modeling has also been explored for hierarchical relations between pathology reports and whole-slide images (Xiong et al., 2024). Despite these advances in static settings, how hyperbolic multimodal representations behave under continual learning, where the geometric structure encoding semantic hierarchy must be preserved across sequential tasks, remains unexplored.

## C.2. Continual Learning

Continual learning aims to enable models to sequentially acquire new knowledge while retaining previously learned information (Yu et al., 2024a). The central challenge is mitigating catastrophic forgetting, where learning new tasks degrades performance on earlier ones. Existing approaches can be broadly categorized into three types: (i) *replay-based methods*, which store and rehearse representative samples from previous tasks, such as Gradient Episodic Memory (GEM) (Lopez-Paz & Ranzato, 2017) and Dark Experience Replay (DER) (Buzzega et al., 2020); (ii) *regularization-based methods*, which constrain parameter updates to preserve important weights, exemplified by Elastic Weight Consolidation (EWC) (Kirkpatrick et al., 2017); and (iii) *parameter isolation methods*, which allocate distinct model components to different tasks to prevent interference, such as Progressive Neural Networks (Rusu et al., 2016), PackNet (Mallya & Lazebnik, 2018), and Piggyback (Mallya et al., 2018).

Recent work has extended continual learning to vision-language models, where preserving cross-modal alignment poses additional challenges beyond single-modality forgetting (Ni et al., 2023). Ni et al. (Ni et al., 2023) identified *Spatial Disorder*, which comprises intra-modal rotation and inter-modal deviation, as a key failure mode in continual CLIP training. Subsequent methods have explored knowledge distillation to retain zero-shot capabilities (Zheng et al., 2023), mixture-of-experts adapters for parameter-efficient adaptation (Yu et al., 2024b), and contrastive regularization such as C-FLAT (Bian et al., 2024) to align feature representations across tasks. While recent work has explored hyperbolic geometry for continual classification (Ayoughi et al., 2025; Doan et al., 2024), these methods focus on single-modal image classification and do not address cross-modal geometric preservation. In contrast, our work targets hyperbolic multimodal representations, where semantic hierarchy encoded through radial coordinates and entailment cones requires geometry-aware continual learning strategies.

# D. Justifications of Problem Statement

The motivation for this investigation stems from the fundamental challenge in continual learning scenarios where new knowledge must be integrated without catastrophic forgetting of historical information. In hyperbolic spaces, semantic knowledge is encoded through geometric relationships, making the preservation of these geometric structures paramount for maintaining learned representations. That is to maintain the **semantic similarity** and the **semantic specificity (hierarchy)** (Desai et al., 2023). Recall the three preservation goals:

**(P1) Intra-modal preservation**: $\mathbf{Z}_{t-1}^{m,t} \mathbf{G} (\mathbf{Z}_{t-1}^{m,t})^\top = \mathbf{Z}_{t-1}^{m,t-1} \mathbf{G} (\mathbf{Z}_{t-1}^{m,t-1})^\top, \forall m \in \{1, 2, \ldots, M\}$;

**(P2) Inter-modal preservation**: $\mathbf{Z}_{t-1}^{m,t} \mathbf{G} (\mathbf{Z}_{t-1}^{m',t})^\top = \mathbf{Z}_{t-1}^{m,t-1} \mathbf{G} (\mathbf{Z}_{t-1}^{m',t-1})^\top, \forall m, m' \in \{1, 2, \ldots, M\}, m \neq m'$;

**(P3) Hierarchical preservation**: $\|(\mathbf{z}_{t-1}^{m,t})_{[1:d]}\| = \|(\mathbf{z}_{t-1}^{m,t-1})_{[1:d]}\|,, \forall m \in \{1, 2, \ldots, M\}$.

*Table 3.* Correspondence between semantic preservation goals and geometric invariants in hyperbolic space.

| Critical Semantic Properties | Geometric Foundation | Preservation Conditions |
|---|---|---|
| **− Primary Goals** | | |
| Semantic Similarity | $d_\mathcal{L}(\mathbf{x}, \mathbf{y}) \propto \langle \mathbf{x}, \mathbf{y} \rangle_\mathcal{L}$ | **(P1)**, **(P2)** |
| Semantic Specificity (Hierarchy) | $d_\mathcal{L}(\mathbf{x}, \mathbf{o}) \propto \|\mathbf{x}_{[1:d]}\|$ | **(P3)** |
| **− Consequence** (Proposition 2) | | |
| Partial Order Structure | $\mathrm{aper}(\mathbf{x}), \mathrm{ext}(\mathbf{x}, \mathbf{y})$ | *Automatically ensured* |

As shown in Table 3, the preservation goals *directly target the geometric foundations* of hyperbolic semantic encoding.

- **Semantic Similarity:** (**P1**) and (**P2**) preserve semantic similarity through inner product invariance. Since Lorentzian distances (Equation (19)) depend monotonically on inner products $\langle \mathbf{x}, \mathbf{y} \rangle_\mathcal{L}$, preserving these inner products ensures that semantic similarity remain invariant.

- **Semantic Hierarchy:** (**P3**) preserves semantic specificity through spatial norm invariance. The induced norm (Definition 4) indicates that distance to origin depends uniquely on spatial norm $\|\mathbf{x}_{[1:d]}\|$. Larger spatial norms position concepts farther from the generic origin, encoding greater specificity in the hierarchical structure.

---

**Proposition 2** (Concepts Partial Order Invariance). *Let $\mathbf{z}_i^m, \mathbf{z}_i^{m'} \in \mathbb{H}^d$ be hyperbolic embeddings. If the spatial norms $\|(\mathbf{z}_i^m)_{[1:d]}\|$ and Lorentzian inner products $\langle \mathbf{z}_i^m, \mathbf{z}_i^{m'} \rangle_\mathcal{L}$ are preserved, then the entailment relationship $\mathbf{z}_i^m \sqsupseteq \mathbf{z}_i^{m'}$ remains invariant.*

---

*Proof.* The entailment relationship is determined by the condition $\mathcal{L}_{\mathrm{entail}}(\mathbf{z}_i^m, \mathbf{z}_i^{m'}) = 0$, which occurs when:

$$\mathrm{ext}(\mathbf{z}_i^m, \mathbf{z}_i^{m'}) \leq \mathrm{aper}(\mathbf{z}_i^m)$$

From the given definitions:

$$\mathrm{aper}(\mathbf{z}_i^m) = \sin^{-1}\left(\frac{2K}{\|(\mathbf{z}_i^m)_{[1:d]}\|}\right) \tag{22}$$

$$\mathrm{ext}(\mathbf{z}_i^m, \mathbf{z}_i^{m'}) = \cos^{-1}\left(\frac{(\mathbf{z}_i^{m'})_0 + (\mathbf{z}_i^m)_0 \langle \mathbf{z}_i^m, \mathbf{z}_i^{m'} \rangle}{\|(\mathbf{z}_i^m)_{[1:d]}\| \sqrt{(\langle \mathbf{z}_i^m, \mathbf{z}_i^{m'} \rangle)^2 - 1}}\right) \tag{23}$$

Since $\mathrm{aper}(\mathbf{z}_i^m)$ depends solely on $\|(\mathbf{z}_i^m)_{[1:d]}\|$ and $\mathrm{ext}(\mathbf{z}_i^m, \mathbf{z}_i^{m'})$ depends on both $\|(\mathbf{z}_i^m)_{[1:d]}\|$ and $\langle \mathbf{z}_i^m, \mathbf{z}_i^{m'} \rangle$, preserving these geometric quantities ensures that the entailment condition remains unchanged. Therefore, the partial order structure is invariant under geometric preservation. □

# E. Supplementary for Theoretical Characterization

**Definition 8** (Joint Embedding Matrix). *Given $N$ paired samples from task $t \in \mathcal{T}$ processed at stage $s \in \mathcal{S}$ with modalities $m_1, m_2$, and individual embedding matrices $\mathbf{Z}_t^{m,s} \in \mathbb{R}^{N \times (d+1)}$ for each modality $m \in \{m_1, m_2\}$. Let $\mathbf{z}_{t,i}^{m,s}, \mathbf{z}_{t,i}^{m,s'} \in \mathbb{H}^d$ for all $i = 1, \ldots, N$ and $m \in \{m_1, m_2\}$. the joint embedding matrix is defined as:*

$$\mathbf{Z}_{t,\mathrm{all}}^s := \begin{pmatrix} \mathbf{Z}_t^{m_1,s} \\ \mathbf{Z}_t^{m_2,s} \end{pmatrix} \in \mathbb{R}^{2N \times (d+1)},$$

*where the vertical concatenation preserves the correspondence between paired samples across modalities.*

**Definition 9** (Extended Lorentz Gram Matrix). *Given the joint embedding matrix $\mathbf{Z}_{t,\mathrm{all}}^s \in \mathbb{R}^{2N \times (d+1)}$ and the Lorentz metric tensor $\mathbf{G} := \mathrm{diag}(-1, 1, \ldots, 1) \in \mathbb{R}^{(d+1) \times (d+1)}$, we define the extended Lorentz Gram matrix as:*

$$\mathbf{G}_{t,\mathrm{ext}}^s := \mathbf{Z}_{t,\mathrm{all}}^s \mathbf{G} (\mathbf{Z}_{t,\mathrm{all}}^s)^\top \tag{24}$$

$$= \begin{pmatrix} \mathbf{Z}_t^{m_1,s} \mathbf{G} (\mathbf{Z}_t^{m_1,s})^\top & \mathbf{Z}_t^{m_1,s} \mathbf{G} (\mathbf{Z}_t^{m_2,s})^\top \\ \mathbf{Z}_t^{m_2,s} \mathbf{G} (\mathbf{Z}_t^{m_1,s})^\top & \mathbf{Z}_t^{m_2,s} \mathbf{G} (\mathbf{Z}_t^{m_2,s})^\top \end{pmatrix} \in \mathbb{R}^{2N \times 2N}. \tag{25}$$

*The diagonal blocks encode intra-modal Lorentzian inner products and the off-diagonal blocks capture cross-modal geometric relationships in the hyperbolic space.*

**Definition 10** (Lorentz Boost). *A Lorentz boost represents relative motion at constant velocity without any rotation of the spatial coordinate axes. Given a velocity vector $\mathbf{v} \in \mathbb{R}^n$ (in units of the speed of light), where $\|\mathbf{v}\| < 1$ and $\gamma = \frac{1}{\sqrt{1-\|\mathbf{v}\|^2}}$, the Lorentz boost matrix is expressed as:*

$$\mathbf{B} = \begin{bmatrix} \gamma & -\gamma \mathbf{v}^\top \\ -\gamma \mathbf{v} & I + \frac{\gamma^2}{1+\gamma} \mathbf{v} \mathbf{v}^\top \end{bmatrix}$$

**Definition 11** (Lorentz Rotation). *A Lorentz rotation refers to the rotation of the spatial coordinates. The corresponding Lorentz rotation matrices are given by:*

$$\mathbf{R} = \begin{bmatrix} 1^\top & 0^\top \\ 0^\top & \tilde{\mathbf{R}} \end{bmatrix}, \quad \text{where } \tilde{\mathbf{R}}^\top \tilde{\mathbf{R}} = I \text{ and } \det(\tilde{\mathbf{R}}) = 1,$$

*i.e., $\tilde{\mathbf{R}} \in SO(n)$ is a special orthogonal matrix.*

### E.1. Proof of Geometric Characterization

**Lemma 1** (Witt's Extension Theorem (Lam, 2005)). *Let $(\mathcal{V}, Q)$ be a non-degenerate quadratic space, where $\mathcal{V}$ is a vector space and $Q : \mathcal{V} \times \mathcal{V} \to \mathbb{R}$ is a non-degenerate quadratic form. If $\phi : \mathcal{U} \to \mathcal{V}'$ is an isometry between subspaces $\mathcal{U}, \mathcal{V}' \subset \mathcal{V}$, then $\phi$ extends to an isometry of the entire space $\mathcal{V}$.*

**Remark 7.** *The matrix setting of Lemma 1 can be rewritten as: given matrices $\mathbf{Z}_1, \mathbf{Z}_2 \in \mathbb{R}^{m \times n}$ and a non-degenerate symmetric matrix $\mathbf{Q} \in \mathbb{R}^{n \times n}$ satisfying $\mathbf{Z}_1 \mathbf{Q} \mathbf{Z}_1^\top = \mathbf{Z}_2 \mathbf{Q} \mathbf{Z}_2^\top$, there exists an orthogonal transformation $\mathbf{T} \in O(\mathbf{Q})$ such that $\mathbf{Z}_2 = \mathbf{Z}_1 \mathbf{T}^\top$, where $O(\mathbf{Q}) := \{\mathbf{T} \in \mathbb{R}^{n \times n} : \mathbf{T}^\top \mathbf{Q} \mathbf{T} = \mathbf{Q}\}$.*

**Lemma 2** (Hyperboloid Isometries are Proper Orthochronous). *Let $\mathbf{L} \in O(1, d)$ be an isometry that maps the hyperbolic space $\mathbb{H}^d := \{\mathbf{x} \in \mathbb{R}^{d+1} : \mathbf{x}^\top \mathbf{G} \mathbf{x} = -1, x_0 > 0\}$ to itself. Then $\mathbf{L} \in \mathrm{SO}^+(1, d)$.*

*Proof.* Since $\mathbf{L}$ maps $\mathbb{H}^d$ to itself, for any $\mathbf{z} = (z_0, z_1, \ldots, z_d)^\top \in \mathbb{H}^d$ with $z_0 > 0$ and $\mathbf{z}^\top \mathbf{G} \mathbf{z} = -1$, we have $\mathbf{L}\mathbf{z} \in \mathbb{H}^d$, which requires $(\mathbf{L}\mathbf{z})_0 = L_{00} z_0 + \sum_{i=1}^d L_{0i} z_i > 0$ and $(\mathbf{L}\mathbf{z})^\top \mathbf{G} (\mathbf{L}\mathbf{z}) = -1$

First, consider any $\mathbf{z} \in \mathbb{H}^d$. Since $\mathbf{z}^\top \mathbf{G} \mathbf{z} = -1$, we have $z_0^2 = 1 + \sum_{i=1}^d z_i^2 \geq 1$, so $z_0 \geq 1$ (using $z_0 > 0$). For the constraint $(\mathbf{L}\mathbf{z})_0 > 0$ to hold for all such $\mathbf{z}$, consider the limiting case where spatial components approach zero. Taking $\mathbf{z} = (1, 0, \ldots, 0)^\top \in \mathbb{H}^d$, we get $(\mathbf{L}\mathbf{z})_0 = L_{00} > 0$.

From $\mathbf{L} \in O(1, d)$, we have $\mathbf{L}^\top \mathbf{G} \mathbf{L} = \mathbf{G}$. Comparing the (0,0) entries:

$$(\mathbf{L}^\top \mathbf{G} \mathbf{L})_{00} = -L_{00}^2 + \sum_{i=1}^{d} L_{i0}^2 = -1 \tag{26}$$

which rearranges to $L_{00}^2 = 1 + \sum_{i=1}^{d} L_{i0}^2 \geq 1$. Since $L_{00} > 0$, we get $L_{00} \geq 1$.

Next, $\mathbf{L}$ maps the connected manifold $\mathbb{H}^d$ to itself and preserves its orientation as a hypersurface in $\mathbb{R}^{d+1}$. Therefore, we have $\det(\mathbf{L}) = +1$. Finally, combining $\mathbf{L} \in O(1, d)$, $\det(\mathbf{L}) = +1$, and $L_{00} \geq 1$, we conclude $\mathbf{L} \in \mathrm{SO}^+(1, d)$. □

---

**Theorem 2** (Lorentzian Isometry Uniqueness). *Let* $\mathbf{Z}_{t,\mathrm{all}}^s, \mathbf{Z}_{t',\mathrm{all}}^{s'} \in \mathbb{R}^{2N \times (d+1)}$ *be two joint embedding matrices. If the following conditions hold: (1) Gram matrix equivalence:* $\mathbf{G}_{t,\mathrm{ext}}^s = \mathbf{G}_{t',\mathrm{ext}}^{s'}$, *where* $\mathbf{G}_{t,\mathrm{ext}}^s = \mathbf{Z}_{t,\mathrm{all}}^s \mathbf{G} (\mathbf{Z}_{t,\mathrm{all}}^s)^\top$; *(2) Non-degeneracy:* $\mathrm{rank}(\mathbf{Z}_{t,\mathrm{all}}^s) = \mathrm{rank}(\mathbf{Z}_{t',\mathrm{all}}^{s'}) = d + 1$. *Then there exists a **unique** $\mathbf{L} \in \mathrm{SO}^+(1, d)$ such that:* $\mathbf{Z}_{t',\mathrm{all}}^{s'} = \mathbf{Z}_{t,\mathrm{all}}^s \mathbf{L}^\top$, *where* $\mathrm{SO}^+(1, d) := \{\mathbf{L} \in \mathbb{R}^{(d+1) \times (d+1)} : \mathbf{L}^\top \mathbf{G} \mathbf{L} = \mathbf{G}, \det(\mathbf{L}) = 1, L_{00} \geq 1\}$ *is the proper orthochronous Lorentz group.*

---

*Proof.* The proof established follows: (i) the existence of an isometry between the row vectors, (ii) its extension to the whole space (i.e., a global isometry), (iii) restriction of the resulting transformation to the orthochronous Lorentz group $\mathrm{SO}^+(1, d)$, and (iv) uniqueness.

By condition (2), both $\mathbf{Z}_{t,\mathrm{all}}^s$ and $\mathbf{Z}_{t',\mathrm{all}}^{s'}$ have full rank $(d+1)$. Hence, their rows span the entire Lorentzian space $\mathbb{R}^{1,d}$. Define a correspondence between row vectors

$$\phi : \mathbf{z}_{t,i}^{m,s} \mapsto \mathbf{z}_{t',i}^{m,s'} \quad (1 \leq i \leq N, \, m \in \{m_1, m_2\}).$$

Condition (1) guarantees that $\phi$ preserves the Lorentzian inner product:

$$\langle \mathbf{z}_{t,i}^{m,s}, \mathbf{z}_{t,j}^{m',s} \rangle_{\mathcal{L}} = \langle \mathbf{z}_{t',i}^{m,s'}, \mathbf{z}_{t',j}^{m',s'} \rangle_{\mathcal{L}}, \quad \forall i, j, m, m'.$$

Thus $\phi$ is an *isometry* between two generating sets of $\mathbb{R}^{1,d}$.

Since the space $\mathbb{R}^{d+1}$ is non-degenerate, Lemma 1 ensures that $\phi$ extends uniquely to a *global isometry* of the ambient quadratic space (i.e., any isometry between subspaces of a non-degenerate quadratic space can be extended to an isometry of the entire space). Consequently, there exists $\mathbf{L} \in O(1, d)$ such that

$$\mathbf{Z}_{t',\mathrm{all}}^{s'} = \mathbf{Z}_{t,\mathrm{all}}^s \mathbf{L}^\top.$$

It remains to identify the proper subgroup. The orthogonal group $O(1, d)$ consists of four connected components, classified by determinant sign and preservation or reversal of time orientation.

Next, according to the definition of joint embedding matrices (Definition 8), $\mathbf{z}_i^{m,s}, \mathbf{z}_i^{m,s'} \in \mathbb{H}^d$ for all $i = 1, \ldots, N$ and $m \in \{m_1, m_2\}$. This ensures that all embeddings lie on the forward sheet $\mathbb{H}^d$ (Definition 3) and $\mathbf{L}$ transform $\mathbb{H}^d$ to itself, so $\mathbf{L} \in \mathrm{SO}^+(1, d)$ (Lemma 2).

Finally, uniqueness follows from full rank. Suppose both $\mathbf{L}_1, \mathbf{L}_2 \in \mathrm{SO}^+(1, d)$ satisfy the required relation, i.e., $\mathbf{Z}_{t',\mathrm{all}}^{s'} = \mathbf{Z}_{t,\mathrm{all}}^s \mathbf{L}_1^\top = \mathbf{Z}_{t,\mathrm{all}}^s \mathbf{L}_2^\top$. Then

$$\mathbf{Z}_{t,\mathrm{all}}^s (\mathbf{L}_1^\top - \mathbf{L}_2^\top) = 0.$$

Since $\mathbf{Z}_{t,\mathrm{all}}^s$ has full rank $(d+1)$, it follows that $\mathbf{L}_1 = \mathbf{L}_2$. This proves the uniqueness claim.

Combining the existence, subgroup restriction, and uniqueness arguments, we obtain the desired result and complete the proof. □

**Corollary 3.** *Under the conditions of Theorem 2, the unique Lorentz transformation $\mathbf{L} \in \mathrm{SO}^+(1, d)$ acts identically on both modalities:*

$$\mathbf{Z}_{t'}^{m_1,s'} = \mathbf{Z}_t^{m_1,s} \mathbf{L}^\top \quad \text{and} \quad \mathbf{Z}_{t'}^{m_2,s'} = \mathbf{Z}_t^{m_2,s} \mathbf{L}^\top.$$

*In particular, if $\mathbf{L}_1, \mathbf{L}_2 \in \mathrm{SO}^+(1, d)$ are transformations such that $\mathbf{Z}_{t'}^{m_i,s'} = \mathbf{Z}_t^{m_i,s} \mathbf{L}_i^\top$ for $i \in \{1, 2\}$, then $\mathbf{L}_1 = \mathbf{L}_2$.*

*Proof.* We prove by contradiction. Suppose there exist distinct transformations $\mathbf{L}_1, \mathbf{L}_2 \in \mathrm{SO}^+(1, d)$ with $\mathbf{L}_1 \neq \mathbf{L}_2$ such that:

$$\mathbf{Z}_{t'}^{m_1,s'} = \mathbf{Z}_t^{m_1,s} \mathbf{L}_1^\top \tag{27}$$

$$\mathbf{Z}_{t'}^{m_2,s'} = \mathbf{Z}_t^{m_2,s} \mathbf{L}_2^\top \tag{28}$$

Then the joint embedding transformation becomes:

$$\mathbf{Z}_{t',\mathrm{all}}^{s'} = \begin{pmatrix} \mathbf{Z}_{t'}^{m_1,s'} \\ \mathbf{Z}_{t'}^{m_2,s'} \end{pmatrix} = \begin{pmatrix} \mathbf{Z}_t^{m_1,s} \mathbf{L}_1^\top \\ \mathbf{Z}_t^{m_2,s} \mathbf{L}_2^\top \end{pmatrix}$$

However, this contradicts Theorem 2, which guarantees the existence of a *unique* $\mathbf{L} \in \mathrm{SO}^+(1, d)$ such that:

$$\mathbf{Z}_{t',\mathrm{all}}^{s'} = \mathbf{Z}_{t,\mathrm{all}}^s \mathbf{L}^\top = \begin{pmatrix} \mathbf{Z}_t^{m_1,s} \mathbf{L}^\top \\ \mathbf{Z}_t^{m_2,s} \mathbf{L}^\top \end{pmatrix}$$

This implies that the transformation matrices for the two modalities must be the same to maintain consistency across the entire embedding. Specifically, the assumption that $\mathbf{L}_1 \neq \mathbf{L}_2$ leads to a contradiction, as it violates the invariance of the cross-modal blocks. These cross-modal blocks, represented by the off-diagonal blocks of the extended Gram matrix, would no longer remain consistent (i.e., $\mathbf{Z}_t^{m_1,s} \mathbf{L}_1^\top \mathbf{G} \mathbf{L}_2 \mathbf{Z}_t^{m_2,s} \neq \mathbf{Z}_{t'}^{m_1,s'} \mathbf{G} \mathbf{Z}_{t'}^{m_2,s'}$), *breaking the requirement that the inter-modal relationships are preserved.* Hence, any valid transformation matrices for the individual modalities must be identical: $\mathbf{L}_1 = \mathbf{L}_2$.

$\square$

**Remark 8.** *Based on Definition 9, the preservation conditions (P1) and (P2) can be equivalently expressed as $\mathbf{G}_{t-1,\mathrm{ext}}^t = \mathbf{G}_{t-1,\mathrm{ext}}^{t-1}$, where $\mathbf{G}_{t-1,\mathrm{ext}}^t$ represents the extended Lorentz Gram matrix for task $t-1$ data processed at the stage $t$. Theorem 2 and Corollary 3 indicate that **this preservation uniquely determines a single transformation $\mathbf{L} \in \mathrm{SO}^+(1, d)$ applying uniformly to both modalities.***

**Lemma 3** (Generation of $\mathrm{SO}^+(1, d)$ (Moretti, 2002))**.** *Every element of $\mathrm{SO}^+(1, d)$ can be expressed as a finite composition of Lorentz rotations and boosts.*

**Remark 9.** *Both the Lorentz boost and the Lorentz rotation are linear transformations defined directly within the Lorentz manifold. Specifically, for any $\mathbf{x} \in \mathbb{H}^d$, the transformations $\mathbf{B}\mathbf{x}$ and $\mathbf{R}\mathbf{x}$ both result in vectors in $\mathbb{H}^d$.*

---

**Theorem 3** (Boost Elimination, Restatement)**.** *Let $\mathbf{L} \in \mathrm{SO}^+(1, d)$ satisfy $\mathbf{z}_{t-1}^{m,t} = \mathbf{L}\mathbf{z}_{t-1}^{m,t-1}$ for embeddings from task $t-1$. If condition (P3) holds:*

$$\|(\mathbf{z}_{t-1}^{m,t})_{[1:d]}\| = \|(\mathbf{z}_{t-1}^{m,t-1})_{[1:d]}\|$$

*for all embeddings, then $\mathbf{L}$ must be a pure spatial rotation:*

$$\mathbf{L} = \begin{bmatrix} 1 & \mathbf{0}^\top \\ \mathbf{0} & \tilde{\mathbf{R}} \end{bmatrix}, \quad \tilde{\mathbf{R}} \in \mathrm{SO}(d).$$

---

*Proof.* By Lemma 3, write $\mathbf{L} = \mathbf{R}\mathbf{B}$ where $\mathbf{R}$ is a spatial rotation and $\mathbf{B}$ is a boost.

For any embedding $\mathbf{z} = [z_0, \mathbf{z}_{[1:d]}]^\top$, according to Definition 10 we have:

$$(\mathbf{L}\mathbf{z})_{[1:d]} = (\mathbf{R}\mathbf{B}\mathbf{z})_{[1:d]} = \tilde{\mathbf{R}}(\mathbf{B}\mathbf{z})_{[1:d]} = \tilde{\mathbf{R}}\left(-\gamma \mathbf{v} z_0 + \left(\mathbf{I} + \frac{\gamma^2}{1+\gamma}\mathbf{v}\mathbf{v}^\top\right)\mathbf{z}_{[1:d]}\right).$$

Since $\tilde{\mathbf{R}}$ preserves norms, condition (P3) requires:

$$\left\|-\gamma \mathbf{v} z_0 + \left(\mathbf{I} + \frac{\gamma^2}{1+\gamma}\mathbf{v}\mathbf{v}^\top\right)\mathbf{z}_{[1:d]}\right\| = \|\mathbf{z}_{[1:d]}\|.$$

**Case 1**: If $\mathbf{v} = \mathbf{0}$, then $\gamma = 1$. Hence, $\mathbf{B} = \mathbf{I}$ and the condition trivially holds.

**Case 2**: If $\mathbf{v} \neq \mathbf{0}$, consider the specific embedding $\mathbf{z} = [1, \mathbf{0}]^\top$ (i.e., origin $\mathbf{o}$). Then:

$$
\begin{aligned}
(\mathbf{Bz})_{[1:d]} &= -\gamma \mathbf{v} \cdot 1 + \left( \mathbf{I} + \frac{\gamma^2}{1+\gamma} \mathbf{v}\mathbf{v}^\top \right) \cdot \mathbf{0} \\
&= -\gamma \mathbf{v} + \mathbf{0} \\
&= -\gamma \mathbf{v}.
\end{aligned}
\tag{29}
$$

The spatial norm after transformation is $\| - \gamma \mathbf{v} \| = \gamma \|\mathbf{v}\| > 0$, but the original spatial norm is $\|\mathbf{0}\| = 0$. Therefore, $\|(\mathbf{Bz})_{[1:d]}\| = \gamma \|\mathbf{v}\| \neq 0 = \|\mathbf{z}_{[1:d]}\|$, which violates condition (**P3**).

Therefore, we must have $\mathbf{v} = \mathbf{0}$, implying $\mathbf{B} = \mathbf{I}$ and $\mathbf{L} = \mathbf{R}$. Since this argument applies to any embedding satisfying condition (**P3**), we conclude that $\mathbf{L}$ must be a pure spatial rotation of the form given in the theorem statement.

$\square$

### E.2. Proof of Hierarchy Preservation

This subsection provides technical proofs supporting the analysis in Appendix 4.2. In particular, the following lemma establishes a first-order relationship between spatial and time-like variations of Lorentz embeddings, which is subsequently used to derive the update constraint in Corollary 1.

> **Lemma 4** (First-order variation of the time-like coordinate). *Let* $\mathbf{z} \in \mathbb{R}^{d+1}$ *be a Lorentz embedding satisfying* $\mathbf{z}_{[0]}^2 - \|\mathbf{z}_{[1:d]}\|_2^2 = -\frac{1}{K}$ *with* $\mathbf{z}_{[0]} > 0$. *Then, for a first-order perturbation* $\Delta \mathbf{z}$, *the variation of the time-like coordinate satisfies*
>
> $$
> \Delta \mathbf{z}_{[0]} = \left( \frac{\mathbf{z}_{[1:d]}^\top}{\mathbf{z}_{[0]}} \right) \Delta \mathbf{z}_{[1:d]}.
> \tag{30}
> $$

*Proof.* Consider the Lorentz-hyperboloid constraint

$$
\phi(\mathbf{z}) := \mathbf{z}_{[0]}^2 - \|\mathbf{z}_{[1:d]}\|_2^2 + \frac{1}{K} = 0.
\tag{31}
$$

For a perturbation $\widetilde{\mathbf{z}} = \mathbf{z} + \Delta \mathbf{z}$, a first-order Taylor expansion of $\phi$ at $\mathbf{z}$ yields

$$
\begin{aligned}
\phi(\widetilde{\mathbf{z}}) &= \phi(\mathbf{z}) + \nabla\phi(\mathbf{z})^\top \Delta \mathbf{z} + O(\|\Delta\mathbf{z}\|_2^2) \\
&= \nabla\phi(\mathbf{z})^\top \Delta\mathbf{z} + O(\|\Delta\mathbf{z}\|_2^2).
\end{aligned}
$$

Since $\widetilde{\mathbf{z}}$ remains on the manifold up to first order, i.e., $\phi(\widetilde{\mathbf{z}}) = 0$, the linear term must vanish, i.e.,

$$
\nabla\phi(\mathbf{z})^\top \Delta\mathbf{z} = 0.
\tag{32}
$$

The gradient of $\phi$ is given by $\nabla\phi(\mathbf{z}) = \begin{bmatrix} 2\mathbf{z}_{[0]} \\ -2\mathbf{z}_{[1:d]} \end{bmatrix}$. Substituting into Equation (32) yields $2\mathbf{z}_{[0]}\Delta\mathbf{z}_{[0]} - 2\mathbf{z}_{[1:d]}^\top \Delta\mathbf{z}_{[1:d]} = 0$. Dividing by 2 and using $\mathbf{z}_{[0]} > 0$ gives

$$
\Delta\mathbf{z}_{[0]} = \left( \frac{\mathbf{z}_{[1:d]}^\top}{\mathbf{z}_{[0]}} \right) \Delta\mathbf{z}_{[1:d]},
$$

which proves Equation (30).

$\square$

**Corollary 4** (Updates under hierarchical preservation, Restatement). *Consider an update $\Delta\mathbf{z} = \mathbf{z}_{t-1}^{m,t} - \mathbf{z}_{t-1}^{m,t-1}$ applied to the old-task embedding $\mathbf{z}_{t-1}^{m,t-1}$. If the time-like coordinate is required to be first-order invariant, i.e. (P3), then the induced spatial update must satisfy*

$$\mathbf{z}_{[1:d]}^{\top}\Delta\mathbf{z}_{[1:d]} = 0. \tag{33}$$

*Proof.* Let $\mathbf{z} \in \mathbb{R}^{d+1}$ denote an old-task embedding and $\Delta\mathbf{z}$ the induced update. Condition (P3) enforces first-order invariance of the time-like coordinate, i.e., $\Delta\mathbf{z}_{[0]} = 0$. By Lemma 4, any first-order perturbation on the Lorentz hyperboloid satisfies $\Delta\mathbf{z}_{[0]} = \left(\frac{\mathbf{z}_{[1:d]}^{\top}}{\mathbf{z}_{[0]}}\right)\Delta\mathbf{z}_{[1:d]}$. Combining the two relations yields $\mathbf{z}_{[1:d]}^{\top}\Delta\mathbf{z}_{[1:d]} = 0$, since $\mathbf{z}_{[0]} > 0$. $\qquad\square$

# F. Deviations for Methodology

## F.1. Proof of Block-Wise Admissible Updates

**Proposition 3** (Admissible Parameter Updates, Restatement). *Consider the Lorentz transformation layer in Appendix 2.1, with parameter decomposition $\mathbf{W}^{m,t} = [\,\mathbf{w}_0^{m,t}\quad\mathbf{W}_s^{m,t}\,]$. Let $\mathbf{z}_{t-1}^{m,t} = f(\mathbf{W}^{m,t}; \mathbf{z}_{t-1}^{m,*})$ denote the representation of task $t-1$ data at stage $t$. Assume Theorem 1 holds at stage $t-1$, so that the induced isometry preserves the time-like coordinate and acts as a spatial rotation. For an infinitesimal update $\mathbf{W}^{m,t} = \mathbf{W}^{m,t-1} - \eta\,\Delta\mathbf{W}^m$, $\eta \to 0^+$, and ignoring second-order terms, $\Delta\mathbf{W}^m$ is admissible if and only if there exists $\mathbf{\Omega}^{\top} = -\mathbf{\Omega}$ such that*

$$\mathbf{Z}_{t-1}^{m,*}\big(\Delta\mathbf{W}_s^m\big)^{\top} = \mathbf{Z}_{t-1}^{m,t-1}[1{:}d]\mathbf{\Omega}^{\top}. \tag{34}$$

*Moreover, the invariance of the time-like coordinate implies $\mathbf{Z}_{t-1}^{m,*}\big(\Delta\mathbf{w}_0^m\big) = \mathbf{0}$, which reduces to $\Delta\mathbf{w}_0^m = \mathbf{0}$ under a non-degeneracy condition on $\mathbf{Z}_{t-1}^{m,*}$.*

*Proof.* Consider the old-task representations evaluated at stage $t$:

$$
\begin{aligned}
\mathbf{Z}_{t-1}^{m,t}[1{:}d] &= f_{[1:d]}\big(\mathbf{W}^{m,t},\, \mathbf{Z}_{t-1}^{m,*}\big)\\
&= f_{[1:d]}\big(\mathbf{W}^{m,t-1} - \eta\,\Delta\mathbf{W}^m,\, \mathbf{Z}_{t-1}^{m,*}\big) \tag{35}\\
&= f_{[1:d]}\big(\mathbf{W}^{m,t-1},\, \mathbf{Z}_{t-1}^{m,*}\big) - \eta\,\mathrm{D}_{\mathbf{W}}f_{[1:d]}\big(\mathbf{W}^{m,t-1},\, \mathbf{Z}_{t-1}^{m,*}\big)\big[\Delta\mathbf{W}^m\big] + \mathcal{O}(\eta^2) \tag{36}\\
&= \mathbf{Z}_{t-1}^{m,t-1}[1{:}d] - \eta\,\mathbf{Z}_{t-1}^{m,*}\big(\Delta\mathbf{W}_s^m\big)^{\top} + \mathcal{O}(\eta^2). \tag{37}
\end{aligned}
$$

On the other hand, by Theorem 1, preservation of old-task representations implies the existence of a spatial isometry $\mathbf{R} \in SO(d)$ such that

$$
\begin{aligned}
\mathbf{Z}_{t-1}^{m,t}[1{:}d] &= \mathbf{Z}_{t-1}^{m,t-1}[1{:}d]\mathbf{R}^{\top}, \qquad \mathbf{R} = \mathbf{I} + \eta\,\mathbf{\Omega} + \mathcal{O}(\eta^2), \quad \mathbf{\Omega}^{\top} = -\mathbf{\Omega} \tag{38}\\
&= \mathbf{Z}_{t-1}^{m,t-1}[1{:}d] + \eta\,\mathbf{Z}_{t-1}^{m,t-1}[1{:}d]\mathbf{\Omega}^{\top} + \mathcal{O}(\eta^2). \tag{39}
\end{aligned}
$$

Comparing the $O(\eta)$ terms in Equation (37) and Equation (39) yields

$$\mathbf{Z}_{t-1}^{m,*}\big(\Delta\mathbf{W}_s^m\big)^{\top} = \mathbf{Z}_{t-1}^{m,t-1}[1{:}d]\mathbf{\Omega}^{\top}, \qquad \mathbf{\Omega}^{\top} = -\mathbf{\Omega}. \tag{40}$$

Finally, Theorem 1 further implies invariance of the time-like coordinate. Since the time-like component of the Lorentz transformation layer depends linearly on $\mathbf{w}_0^m$, we have

$$\mathbf{Z}_{t-1}^{m,t}[0] = \mathbf{Z}_{t-1}^{m,t-1}[0] - \eta\,\mathbf{Z}_{t-1}^{m,*}\big(\Delta\mathbf{w}_0^m\big) + \mathcal{O}(\eta^2),$$

which implies

$$\mathbf{Z}_{t-1}^{m,*}\big(\Delta\mathbf{w}_0^m\big) = \mathbf{0}. \tag{41}$$

This completes the proof. $\qquad\square$

### F.2. Proof of Canonical Admissible Update

**Corollary 5** (Canonical Admissible Update, Restatement). *Under the admissibility condition of Proposition 1, consider the admissible update associated with a given unconstrained update direction $\boldsymbol{\delta}_s^m$, while penalizing the induced spatial rotation. Then, in the minimal-rotation regime, the admissible solution satisfies $\boldsymbol{\Omega}^\star = \mathbf{0}$, and admissibility reduces to*

$$\mathbf{Z}_{t-1}^{m,*}\big(\Delta \mathbf{W}_s^m\big)^\top = \mathbf{0}, \qquad \Delta \mathbf{w}_0^m = \mathbf{0}. \tag{42}$$

*The resulting canonical admissible update is given by the null-space projection*

$$\Delta \mathbf{W}_s^m = \boldsymbol{\delta}_s^m\big(\mathbf{I} - \mathbf{P}_{t-1}\big), \qquad \mathbf{P}_{t-1} = (\mathbf{Z}_{t-1}^{m,*})^\top \Big(\mathbf{Z}_{t-1}^{m,*}(\mathbf{Z}_{t-1}^{m,*})^\top\Big)^\dagger \mathbf{Z}_{t-1}^{m,*}. \tag{43}$$

*Proof.* We start from the general admissibility condition in Proposition 1,

$$\mathbf{Z}_{t-1}^{m,*}\big(\Delta \mathbf{W}_s^m\big)^\top = \mathbf{Z}_{t-1}^{m,t-1}[1\!:\!d]\boldsymbol{\Omega}^\top, \qquad \boldsymbol{\Omega}^\top = -\boldsymbol{\Omega}. \tag{44}$$

Let $\boldsymbol{\delta}_s^m$ denote a given unconstrained update direction for the spatial block. We consider the rotation-regularized admissible update defined by

$$\min_{\Delta \mathbf{W}_s^m, \boldsymbol{\Omega}} \frac{1}{2}\big\|\Delta \mathbf{W}_s^m - \boldsymbol{\delta}_s^m\big\|_F^2 + \frac{\lambda}{2}\|\boldsymbol{\Omega}\|_F^2 \quad \text{s.t.} \quad Equation\,(44). \tag{45}$$

Introducing a Lagrange multiplier $\boldsymbol{\Lambda}$, the Lagrangian is

$$\mathcal{L} = \frac{1}{2}\big\|\Delta \mathbf{W}_s^m - \boldsymbol{\delta}_s^m\big\|_F^2 + \frac{\lambda}{2}\|\boldsymbol{\Omega}\|_F^2 \tag{46}$$

$$+ \big\langle \boldsymbol{\Lambda}, \, \mathbf{Z}_{t-1}^{m,*}(\Delta \mathbf{W}_s^m)^\top - \mathbf{Z}_{t-1}^{m,t-1}[1\!:\!d]\boldsymbol{\Omega}^\top \big\rangle. \tag{47}$$

Taking first-order optimality conditions yields

$$\Delta \mathbf{W}_s^m = \boldsymbol{\delta}_s^m - \boldsymbol{\Lambda}^\top \mathbf{Z}_{t-1}^{m,*}, \tag{48}$$

$$\boldsymbol{\Omega} = \frac{1}{\lambda}\,\mathrm{Skew}\big((\mathbf{Z}_{t-1}^{m,t-1}[1\!:\!d])^\top \boldsymbol{\Lambda}\big), \tag{49}$$

where $\mathrm{Skew}(\mathbf{A}) = (\mathbf{A} - \mathbf{A}^\top)/2$. Substituting Equation (48) into the admissibility constraint Equation (44) gives

$$\mathbf{Z}_{t-1}^{m,*}(\mathbf{Z}_{t-1}^{m,*})^\top \boldsymbol{\Lambda} = \mathbf{Z}_{t-1}^{m,*}(\boldsymbol{\delta}_s^m)^\top + \mathcal{O}(\lambda^{-1}). \tag{50}$$

In the minimal-rotation regime $\lambda \to \infty$, the second term vanishes, yielding

$$\boldsymbol{\Lambda} = \big(\mathbf{Z}_{t-1}^{m,*}(\mathbf{Z}_{t-1}^{m,*})^\top\big)^\dagger \mathbf{Z}_{t-1}^{m,*}(\boldsymbol{\delta}_s^m)^\top. \tag{51}$$

Substituting into Equation (48) gives

$$\Delta \mathbf{W}_s^m = \boldsymbol{\delta}_s^m\big(\mathbf{I} - (\mathbf{Z}_{t-1}^{m,*})^\top\big(\mathbf{Z}_{t-1}^{m,*}(\mathbf{Z}_{t-1}^{m,*})^\top\big)^\dagger \mathbf{Z}_{t-1}^{m,*}\big), \tag{52}$$

while Equation (49) implies $\boldsymbol{\Omega}^\star = \mathbf{0}$. Together with Proposition 1, this completes the proof. $\square$

### F.3. Finite-step optimization with AdamW

The derivation in the main text characterizes local admissibility, while practical training applies the projected update repeatedly with a finite learning rate. In our implementation, the projection matrix is fixed within each task stage and every raw spatial update is first mapped to $\boldsymbol{\delta}_s^m(\mathbf{I} - \mathbf{V}\mathbf{V}^\top)$, so the protected old-task subspace is removed before the optimizer step. For Adam-style momentum, this means the first-moment estimate remains in the projected subspace by induction as long as the stage projector is fixed. Adaptive coordinate-wise preconditioning and finite step sizes can introduce higher-order approximation error, so we do not claim exact global preservation for arbitrary trajectories; instead, HMCL suppresses the dominant first-order leakage into old-task directions, leaving only optimizer-induced and higher-order residuals. Empirically, the robustness analysis in Appendix G.3 shows that the method remains effective under AdamW with nonzero weight decay.

# G. Supplementary for Implementation Details

## G.1. Dataset Details and Task Stream

Our continual multimodal learning setting consisting of 15 datasets, covering both image–text classification and cross-modal retrieval tasks.

**Classification datasets.**   The classification benchmarks include CIFAR-10 and CIFAR-100 (Krizhevsky, 2009), Caltech-101 (Fei-Fei et al., 2004), **Food-101** (Bossard et al., 2014), **Flowers-102** (Nilsback & Zisserman, 2008), **EuroSAT** (Helber et al., 2019), **DTD** (Cimpoi et al., 2014), **CLEVR** (Johnson et al., 2016), **Aircraft** (Maji et al., 2013), **MNIST** (LeCun et al., 1998), **PCAM** (Veeling et al., 2018), **SST-2** (Socher et al., 2013), and **Country211** (Radford et al., 2021). These datasets span diverse visual domains and label granularities, ranging from coarse-grained object recognition to fine-grained and scene-level classification. Each dataset is treated as an independent classification task and evaluated using standard image–text classification protocols.

**Retrieval datasets.**   Cross-modal retrieval is evaluated on **COCO** (Lin et al., 2014) and **Flickr30k** (Young et al., 2014), which consist of paired image–text data and are widely used in multimodal retrieval benchmarks. Both datasets support bidirectional retrieval, including image-to-text and text-to-image retrieval.

**Task stream construction.**   All datasets are arranged into a fixed task stream and presented sequentially. At each step, training is performed on the current dataset only, without access to data from previous tasks. Classification and retrieval tasks are interleaved in the task stream to reflect realistic continual multimodal learning scenarios. The same task order is used across all experiments.

**Dataset statistics.**   Key statistics of all datasets, including the number of samples, the number of classes for classification tasks, and the number of image–text pairs for retrieval tasks, are summarized in Table 4.

*Table 4.* Dataset statistics and task types used in the continual multimodal stream.

| Dataset | Task | Classes | Train | Test | Evaluation |
|---|---|---|---|---|---|
| FGVC Aircraft (Maji et al., 2013) | Classification | 100 | 3,334 | 3,333 | Accuracy |
| Caltech-101 (Fei-Fei et al., 2004) | Classification | 102 | 2,448 | 6,084 | Accuracy |
| CIFAR-10 (Krizhevsky, 2009) | Classification | 10 | 45,000 | 10,000 | Accuracy |
| CIFAR-100 (Krizhevsky, 2009) | Classification | 100 | 45,000 | 10,000 | Accuracy |
| CLEVR (Johnson et al., 2016) | Classification | 8 | 4,500 | 5,000 | Accuracy |
| Country211 (Radford et al., 2021) | Classification | 211 | 31,650 | 21,100 | Accuracy |
| DTD (Cimpoi et al., 2014) | Classification | 47 | 1,880 | 1,880 | Accuracy |
| EuroSAT (Helber et al., 2019) | Classification | 10 | 5,000 | 5,000 | Accuracy |
| Flowers-102 (Nilsback & Zisserman, 2008) | Classification | 102 | 1,020 | 6,149 | Accuracy |
| Food-101 (Bossard et al., 2014) | Classification | 101 | 68,175 | 25,250 | Accuracy |
| MNIST (LeCun et al., 1998) | Classification | 10 | 48,000 | 10,000 | Accuracy |
| PCAM (Veeling et al., 2018) | Classification | 2 | 262,144 | 32,768 | Accuracy |
| SST-2 (Socher et al., 2013) | Classification | 2 | 6,920 | 1,821 | Accuracy |
| COCO (Lin et al., 2014) | Retrieval | – | 118,287 | 5,000 | Recall@K |
| Flickr30K (Young et al., 2014) | Retrieval | – | 29,000 | 1,000 | Recall@K |

## G.2. Supplementary Ablation Study

Figure 5 compares the full HMCL model with a variant that removes the block-wise constraint (*w/o block*) across three hyperbolic backbones. While the ablated variant still outperforms the Vanilla baseline, it consistently underperforms the full model in terms of backward transfer (BWT) for both classification and retrieval. This gap is especially pronounced on MERU-L and MERU-B, indicating that block-wise constraints play a central role. These ablation results confirm that explicitly enforcing geometry-aware constraints at the block level is essential for preserving hyperbolic structure during continual learning. Without such constraints, continual updates remain vulnerable to geometric drift, even when trained with hyperbolic losses. We provide additional optimizer robustness and shuffled task-order analyses in Appendix G.3.

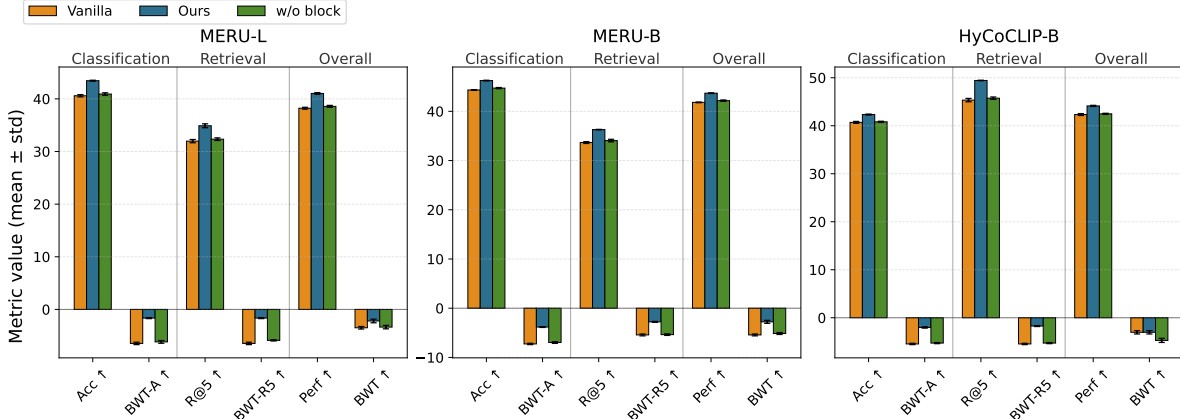

*Figure 5.* **Ablation of block-wise admissible updates.** Results comparing Vanilla, HMCL (*w/ block*), and a variant without block-wise decomposition (*w/o block*). Removing the separation between time-like and space-like updates leads to consistently worse backward transfer, highlighting the importance of block-wise constraints for stability.

*Table 5.* Robustness under AdamW with nonzero weight decay ($10^{-2}$).

| Method | Acc ↑ | BWT$_A$ ↑ | R@5 ↑ | BWT$_{R5}$ ↑ | Performance ↑ | BWT ↑ |
|---|---|---|---|---|---|---|
| Vanilla | 41.22 | -11.41 | 24.67 | -15.41 | 39.01 | -11.94 |
| HMCL (ours) | **45.25** | **-3.31** | **35.50** | **-5.66** | **43.95** | **-3.62** |
| **Δ vs. Vanilla** | **+9.8%** | **+71.0%** | **+43.9%** | **+63.3%** | **+12.7%** | **+69.7%** |

### G.3. Supplementary Robustness and Drift Analyses

We provide the additional analyses requested during rebuttal to clarify the implementation–theory connection and to test whether the observed gains are robust beyond the default optimizer and task order. These experiments are intended as stress tests rather than new assumptions: the AdamW experiment checks optimizer mismatch under nonzero weight decay, the task-order experiment checks sensitivity to shuffled streams, and the drift analysis directly measures whether HMCL reduces both hierarchy-related and relation-related representation changes.

### G.4. Supplementary Materials for Case Study

To further examine the structural advantages of hyperbolic representations in preserving visual–semantic hierarchies, we provide additional geodesic-interpolation case studies on Flickr30K and COCO val2017. For each query image, we interpolate from the image embedding to the [ROOT] node and retrieve the nearest textual item or concept along the geodesic path. This traversal allows us to inspect whether the learned representation moves smoothly from image-specific semantics toward more abstract concepts near the origin. We compare our method, HMCL, with a Vanilla continual fine-tuning baseline.

**Flickr30K case study.** For Flickr30K, we construct a hierarchical text set by extracting nouns and adjectives from the original captions, so that the retrieved items reflect different levels of semantic abstraction. As shown in Figure 6 and Figure 7, cases (1)–(3) perform retrieval over a mixed pool of nouns, adjectives, and captions, while cases (4)–(6) restrict the retrieval pool to captions only. The blue rows correspond to the model state immediately after training on Flickr30K, and the green rows correspond to the model state after subsequent continual-learning tasks. HMCL exhibits fine-grained transitions from image-specific descriptions to more generic concepts and preserves this structure after continual learning, whereas Vanilla is more prone to semantic drift and hierarchical collapse.

**COCO case study.** We further conduct a case study on COCO val2017 to more directly evaluate whether the learned visual–semantic hierarchy remains stable after continual learning. For each COCO image, we first retrieve the nearest caption anchor from the full COCO validation caption pool. We then trace the geodesic path from the image embedding to the origin and retrieve the nearest concepts from a curated three-level vocabulary, where upper rows correspond to more

*Table 6.* Stability under shuffled task orders.

| Order | Backbone | Method | Classification | | Retrieval | | Overall | |
|---|---|---|---|---|---|---|---|---|
| | | | Acc ↑ | BWT$_A$ ↑ | R@5 ↑ | BWT$_{R5}$ ↑ | Performance ↑ | BWT ↑ |
| 1 | MERU-L | Vanilla | 41.00 | -11.00 | 29.67 | -10.94 | 38.40 | -11.00 |
| 1 | MERU-L | HMCL (ours) | **44.09** | **-1.24** | **35.78** | **-4.60** | **42.99** | **-1.68** |
| 1 | MERU-L | Δ **vs. Vanilla** | **+7.5%** | **+88.7%** | **+20.6%** | **+57.9%** | **+12.0%** | **+84.7%** |
| 1 | MERU-B | Vanilla | 43.46 | -10.49 | 25.29 | -17.09 | 40.03 | -10.49 |
| 1 | MERU-B | HMCL (ours) | **45.25** | **-3.31** | **35.50** | **-5.66** | **43.95** | **-3.62** |
| 1 | MERU-B | Δ **vs. Vanilla** | **+4.1%** | **+68.4%** | **+40.4%** | **+66.9%** | **+9.8%** | **+65.5%** |
| 1 | HyCoCLIP-B | Vanilla | 42.49 | -8.32 | 32.28 | -15.13 | 43.59 | -8.32 |
| 1 | HyCoCLIP-B | HMCL (ours) | **44.94** | **-0.38** | **48.42** | **-3.83** | **48.62** | **-0.81** |
| 1 | HyCoCLIP-B | Δ **vs. Vanilla** | **+5.8%** | **+95.4%** | **+50.0%** | **+74.7%** | **+11.5%** | **+90.3%** |
| 2 | MERU-L | Vanilla | 43.83 | -7.84 | 28.43 | -13.47 | 40.73 | -7.84 |
| 2 | MERU-L | HMCL (ours) | **44.42** | **-1.19** | **35.39** | **-4.66** | **43.21** | **-1.65** |
| 2 | MERU-L | Δ **vs. Vanilla** | **+1.3%** | **+84.8%** | **+24.5%** | **+65.4%** | **+6.1%** | **+79.0%** |
| 2 | MERU-B | Vanilla | 43.23 | -9.29 | 22.09 | -20.00 | 39.53 | -9.29 |
| 2 | MERU-B | HMCL (ours) | **45.34** | **-2.45** | **35.68** | **-4.74** | **44.05** | **-2.76** |
| 2 | MERU-B | Δ **vs. Vanilla** | **+4.9%** | **+73.6%** | **+61.5%** | **+76.3%** | **+11.4%** | **+70.3%** |
| 2 | HyCoCLIP-B | Vanilla | 43.32 | -6.38 | 29.99 | -17.53 | 44.23 | -6.38 |
| 2 | HyCoCLIP-B | HMCL (ours) | **45.33** | **-0.15** | **48.66** | **-4.35** | **49.00** | **-0.60** |
| 2 | HyCoCLIP-B | Δ **vs. Vanilla** | **+4.6%** | **+97.6%** | **+62.3%** | **+75.2%** | **+10.8%** | **+90.6%** |

*Table 7.* Summary of quantitative representation-drift analysis. Reductions are averaged over old tasks and compare HMCL against Vanilla.

| Drift metric | What it measures | Reduction |
|---|---|---|
| Radial drift | Changes in hyperbolic radius / time-like hierarchy of old-task embeddings. | 61.0% |
| Angular drift | Within-modality directional distortion after removing magnitude effects. | 57.8% |
| Cross-modal drift | Changes in image–text relational structure for old-task representations. | 73.8% |
| Paired-distance drift | Increase in geodesic distance between matched image–text pairs. | 59.5% |

The reductions are statistically significant across old tasks ($p \leq 3.1 \times 10^{-4}$). The analysis supports the revised interpretation that forgetting includes both hierarchy-related distortion and relation-level drift, rather than being caused solely by radial changes.

image-specific concepts and lower rows approach more abstract concepts near [ROOT].

We compare HMCL and Vanilla at two checkpoints: $T = 6$, immediately after COCO is learned, and $T = 15$, after nine subsequent tasks. The selected cases follow a hard filtering protocol: both models must correctly retrieve the caption and hierarchy anchor at $T = 6$, while at $T = 15$, HMCL must still preserve both, and Vanilla must fail in at least one of them. This protocol isolates cases where both models initially solve the task, but only HMCL maintains the visual–semantic hierarchy after continual learning. The vertical paths in Figure 8–Figure 11 show that HMCL maintains stable semantic anchors and smooth abstraction toward the origin, whereas Vanilla is more likely to exhibit caption drift or hierarchy-anchor collapse after subsequent tasks.

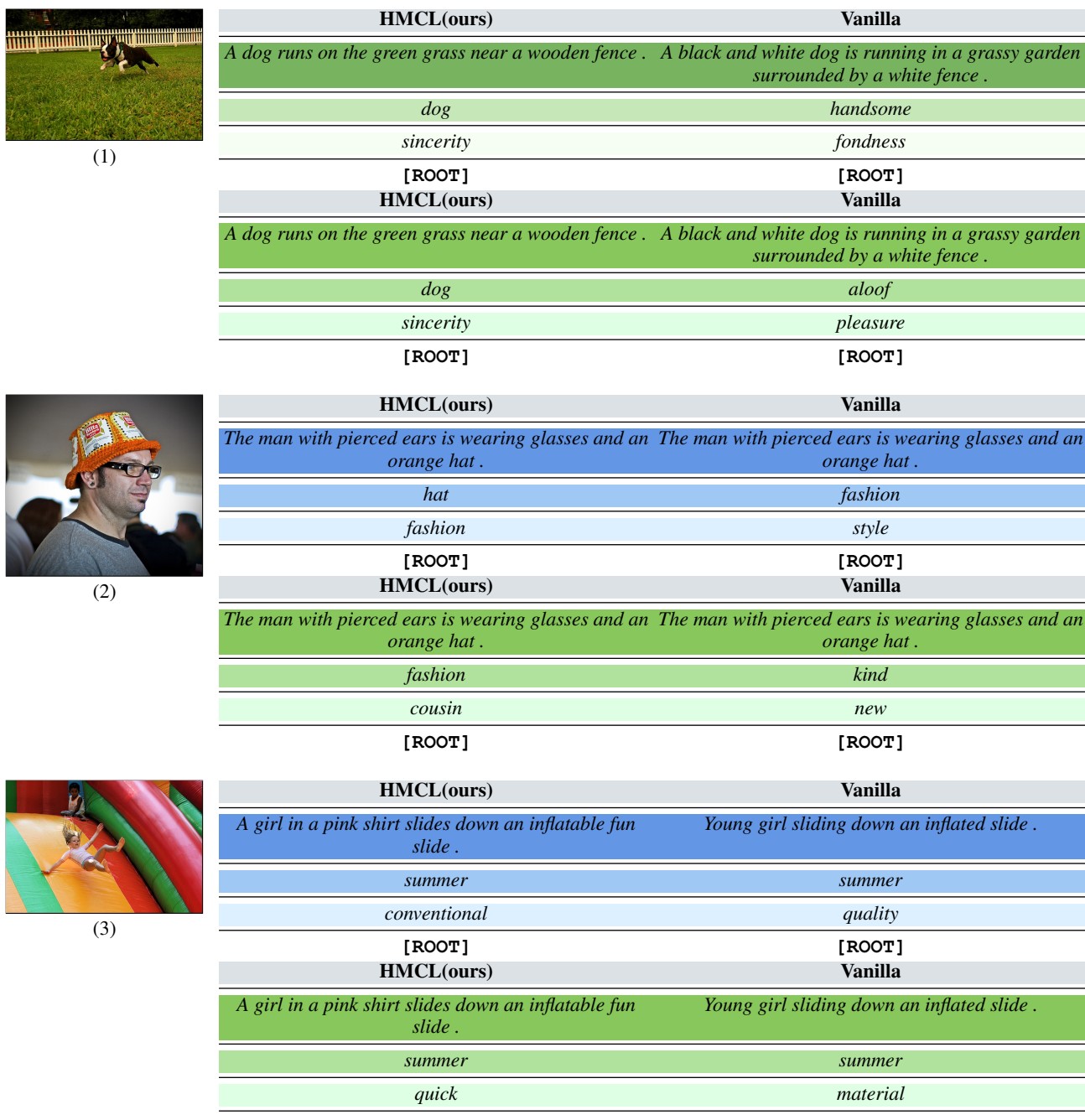

| HMCL(ours) | Vanilla |
|---|---|
| *A dog runs on the green grass near a wooden fence .* | *A black and white dog is running in a grassy garden surrounded by a white fence .* |
| *dog* | *handsome* |
| *sincerity* | *fondness* |
| **[ROOT]** | **[ROOT]** |

| HMCL(ours) | Vanilla |
|---|---|
| *A dog runs on the green grass near a wooden fence .* | *A black and white dog is running in a grassy garden surrounded by a white fence .* |
| *dog* | *aloof* |
| *sincerity* | *pleasure* |
| **[ROOT]** | **[ROOT]** |

| HMCL(ours) | Vanilla |
|---|---|
| *The man with pierced ears is wearing glasses and an orange hat .* | *The man with pierced ears is wearing glasses and an orange hat .* |
| *hat* | *fashion* |
| *fashion* | *style* |
| **[ROOT]** | **[ROOT]** |

| HMCL(ours) | Vanilla |
|---|---|
| *The man with pierced ears is wearing glasses and an orange hat .* | *The man with pierced ears is wearing glasses and an orange hat .* |
| *fashion* | *kind* |
| *cousin* | *new* |
| **[ROOT]** | **[ROOT]** |

| HMCL(ours) | Vanilla |
|---|---|
| *A girl in a pink shirt slides down an inflatable fun slide .* | *Young girl sliding down an inflated slide .* |
| *summer* | *summer* |
| *conventional* | *quality* |
| **[ROOT]** | **[ROOT]** |

| HMCL(ours) | Vanilla |
|---|---|
| *A girl in a pink shirt slides down an inflatable fun slide .* | *Young girl sliding down an inflated slide .* |
| *summer* | *summer* |
| *quick* | *material* |
| **[ROOT]** | **[ROOT]** |

*Figure 6.* Flickr30K geodesic-interpolation case studies, page 1. Cases (1)–(3) retrieve over a mixed pool of captions, nouns, and adjectives. Blue rows show the checkpoint immediately after learning Flickr30K, and green rows show the final checkpoint after subsequent continual-learning tasks.

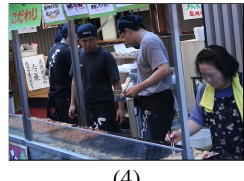

(4)

| HMCL(ours) | Vanilla |
| --- | --- |
| *Numerous Asian people working behind a counter .* | *Numerous Asian people working behind a counter .* |
| *An Asian food market where a woman is picking out food .* | *An Asian food market where a woman is picking out food .* |
| *Employees at a sushi restaurant prepare for dinner time rush .* | *A cyclist wearing a black helmet is riding by some black vans .* |
| **[ROOT]** | **[ROOT]** |
| **HMCL(ours)** | **Vanilla** |
| *Numerous Asian people working behind a counter .* | *Employees at a sushi restaurant prepare for dinner time rush .* |
| *An Asian food market where a woman is picking out food .* | *A cyclist wearing a black helmet is riding by some black vans .* |
| *A cyclist wearing a black helmet is riding by some black vans .* | *Two wet dogs run into the surf at sunset .* |
| **[ROOT]** | **[ROOT]** |

(5)

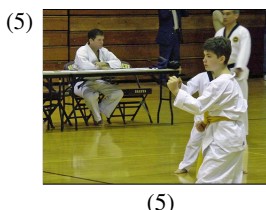

(5)

| HMCL(ours) | Vanilla |
| --- | --- |
| *2 boys in the foreground in a karate competition and coaches in background looking on with another coach sitting at table .* | *A young boy demonstrates karate in a gymnasium .* |
| *A young boy demonstrates karate in a gymnasium .* | *A little girl dressed in yellow splashes in a shallow pool .* |
| *Man standing by a poster of religious beliefs .* | *A child plays at a playground .* |
| **[ROOT]** | **[ROOT]** |
| **HMCL(ours)** | **Vanilla** |
| *2 boys in the foreground in a karate competition and coaches in background looking on with another coach sitting at table .* | *Two people are sitting outdoors on a blanket near a tree .* |
| *A young boy demonstrates karate in a gymnasium .* | *A surfer jumps a wave .* |
| *Man standing by a poster of religious beliefs .* | *A couple enjoying a glass of white wine .* |
| **[ROOT]** | **[ROOT]** |

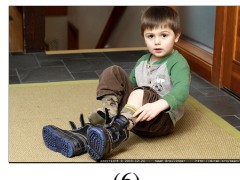

(6)

| HMCL(ours) | Vanilla |
| --- | --- |
| *A young boy is sitting on the floor trying to take off his boots .* | *A woman wearing glasses and an orange dress is standing with another woman wearing glasses and a green dress .* |
| *A young boy puts on his boots at the top of a flight of stairs .* | *Three young men wearing hooded sweatshirts , standing at a bench .* |
| *A young boy demonstrates karate in a gymnasium .* | *A couple enjoying a glass of white wine .* |
| **[ROOT]** | **[ROOT]** |
| **HMCL(ours)** | **Vanilla** |
| *A young boy is sitting on the floor trying to take off his boots .* | *An Indian girl playing a guitar .* |
| *A young boy puts on his boots at the top of a flight of stairs .* | *A couple enjoying a glass of white wine .* |
| *couple enjoying a glass of white wine .* | *Two wet dogs run into the surf at sunset .* |
| **[ROOT]** | **[ROOT]** |

*Figure 7.* Flickr30K geodesic-interpolation case studies, page 2. Cases (4)–(6) restrict retrieval to the caption pool. Blue rows show the checkpoint immediately after learning Flickr30K, and green rows show the final checkpoint after subsequent continual-learning tasks.

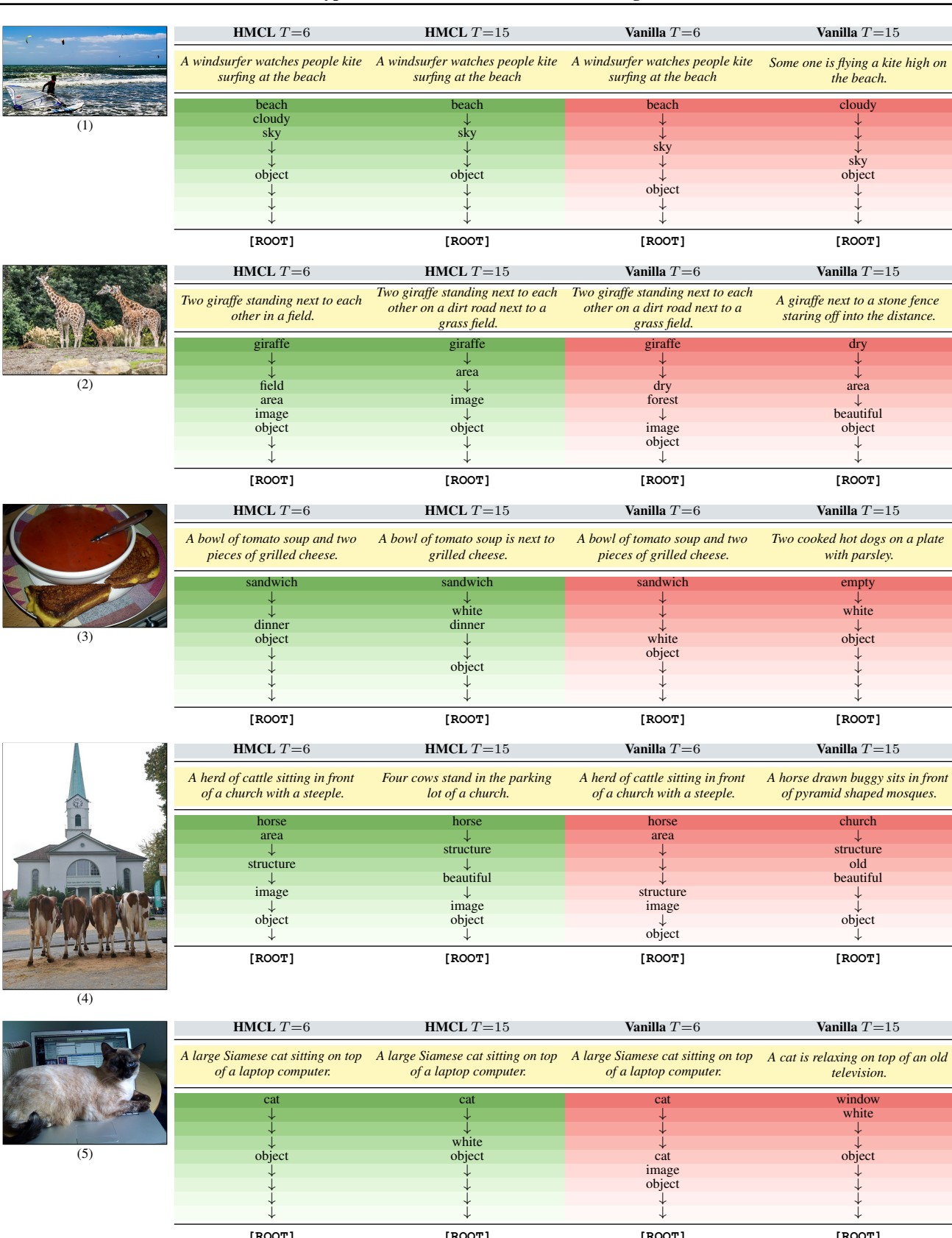

*Figure 8.* COCO geodesic-interpolation case studies, page 1.

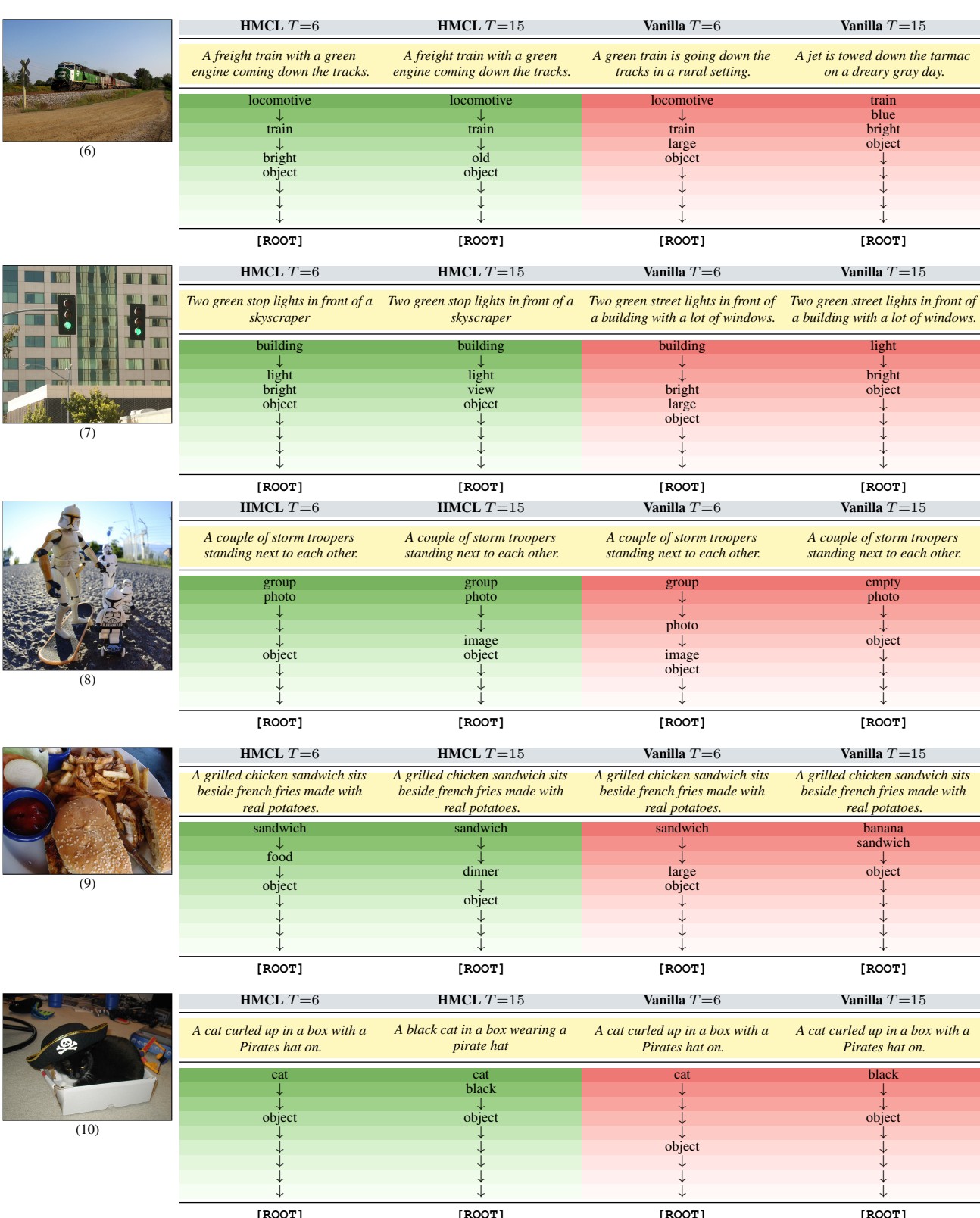

*Figure 9.* COCO geodesic-interpolation case studies, page 2.

# Hyperbolic Multimodal Continual Learning

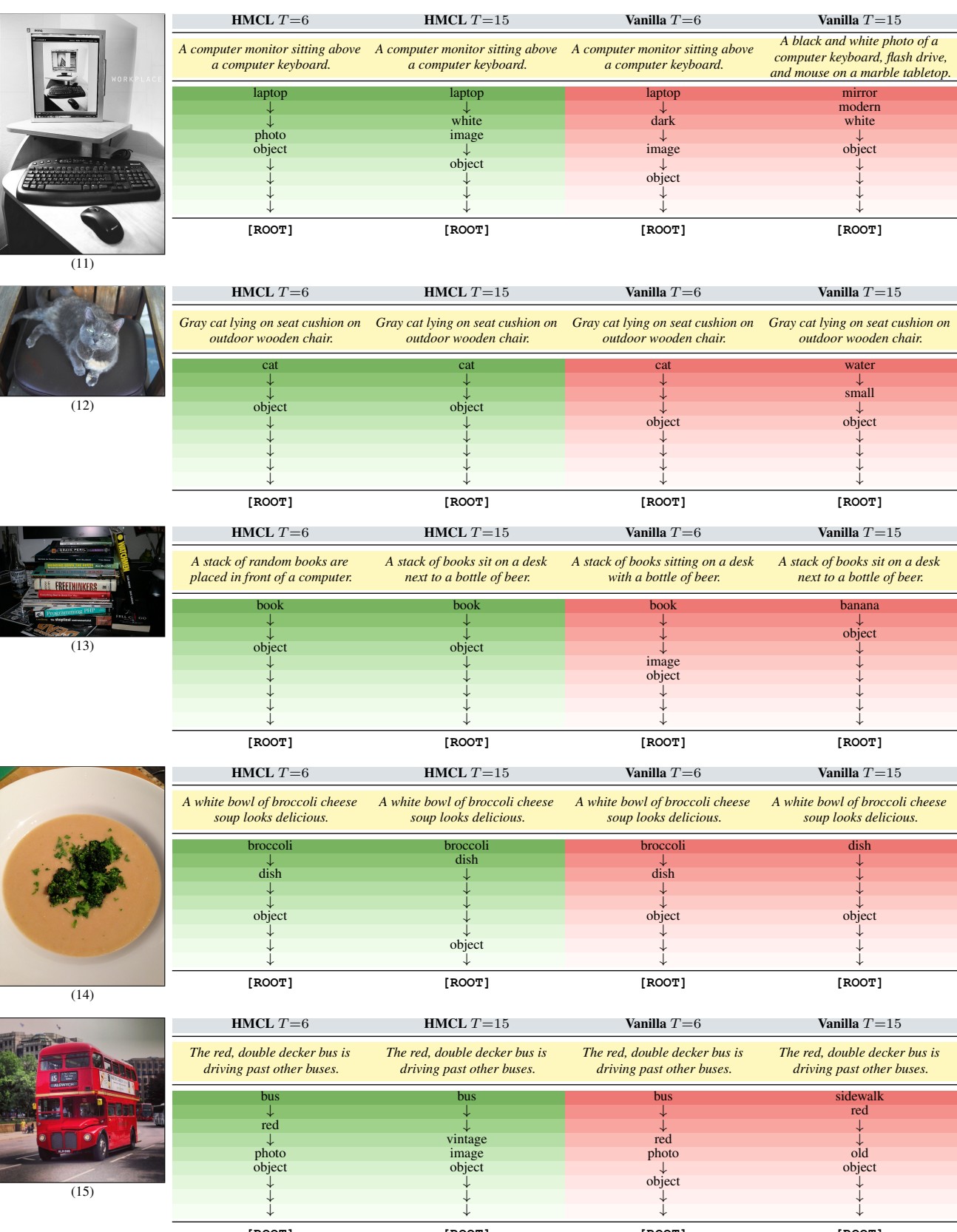

*Figure 10.* COCO geodesic-interpolation case studies, page 3.

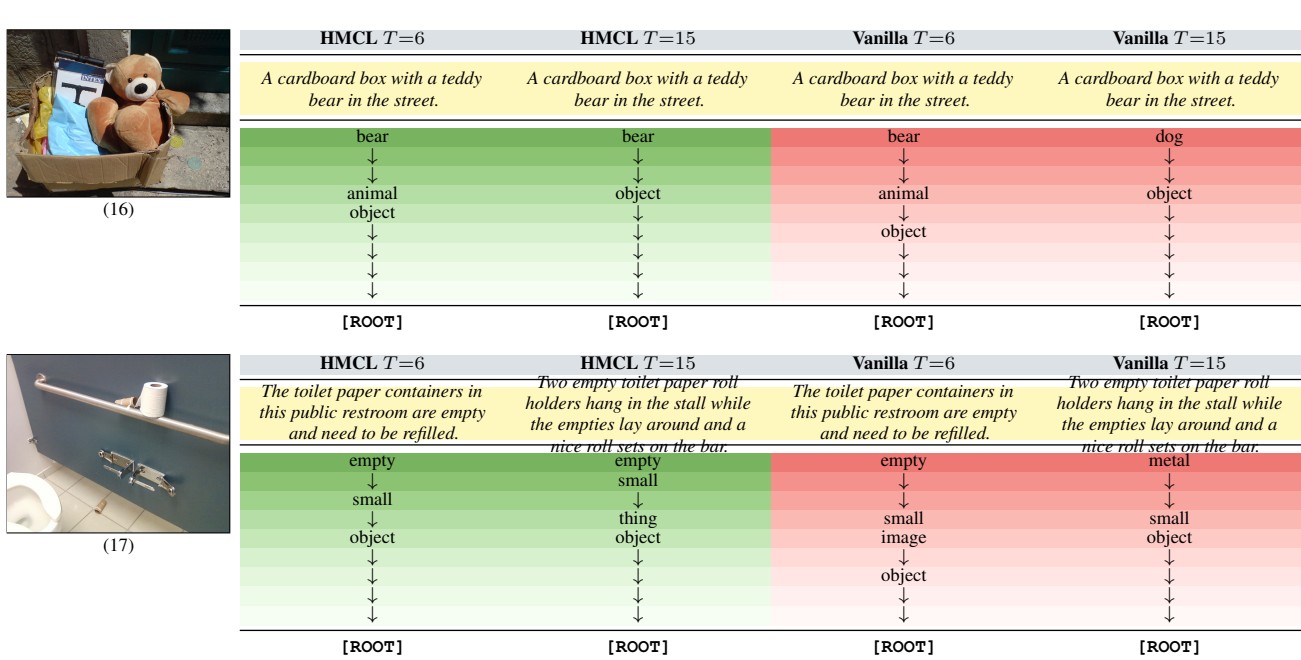

*Figure 11.* COCO geodesic-interpolation case studies, page 4.

