# OpenReview forum: "Hyperbolic Multimodal Continual Learning"
_ICML.cc/2026/Conference — ICML 2026 regular_

### Official Review · Reviewer_GesZ · 2026-03-12

**Soundness:** 3
**Presentation:** 3
**Significance:** 3
**Originality:** 3
**Overall Recommendation:** 5
**Confidence:** 4

**Summary:**

This paper explores continual learning for hyperbolic multimodal representations. The main idea is that forgetting in this context should be examined from a geometric perspective. To preserve prior knowledge, it is required to maintain the similarity structure within each modality, ensure alignment between modalities, and uphold the hierarchical structure represented by the distance from the origin. The paper formalizes these considerations through three preserving conditions (P1)–(P3), proves a characterization based on a shared hyperbolic isometry, and derives first-order admissible update conditions for a Lorentz transformation layer. It introduces HMCL, which zeros out time-like updates and projects spatial updates out of the old-task subspace. Empirically, HMCL is tested on up to 15 sequential multimodal tasks across three datasets: MERU-L, MERU-B, and HyCoCLIP-B. The results demonstrate consistent improvements in average performance and backward transfer compared to Vanilla, EWC, GEM, and C-FLAT methods.

**Compliance With Llm Reviewing Policy:**

Affirmed.

**Final Justification:**

My concerns have been addressed.

**Key Questions For Authors:**

1. Proposition 1 is derived for infinitesimal updates. What evidence supports that the finite-step AdamW procedure used in training still preserves the intended invariants over long task sequences?

2. This paper repeatedly claims that forgetting is dominated by radial distortion. Can you provide a direct quantitative analysis of radial drift vs. inter-modal drift, and correlate these with forgetting metrics?

3. How stable are the conclusions under multiple task orders, especially adversarial or shuffled interleaving of classification and retrieval tasks?

**Limitations:**

No limitations and societal impacts are discussed.

**Strengths And Weaknesses:**

**Strengths**

1. This paper addresses an interesting and novel intersection: continual learning for hyperbolic multimodal models. The geometric framing is coherent and more principled than a purely heuristic anti-forgetting regularizer.

2. The progression from preservation conditions to shared-isometry characterization to first-order admissible updates to the projected update rule is clean and organized.

3. The empirical section is reasonably broad in backbone coverage and task variety, and the reported gains are consistent across all three backbones.

4. The method is lightweight, introducing fewer extra learnable parameters beyond the existing transformation layer and relying only on a projection computed from old-task representations.

**Weaknesses**

1. There is a gap between your theory and algorithm. Theorem 1 gives an exact preservation characterization, and Proposition 1 is stated for an infinitesimal update $\mathbf{W}^{m,t} = \mathbf{W}^{m,t-1} - \eta \Delta \mathbf{W}^m$ with $\eta$ $\rightarrow$ $0^+$. However, the actual training uses AdamW with finite learning rates and multiple epochs per task. This paper does not establish that repeated finite-step projected optimization approximately preserves the same invariants over long task streams, nor does it provide an error accumulation analysis. The theory gives strong geometric motivation, but not a guarantee for the implemented optimizer.

2. Both Theorem 1 and Proposition 1 depend on non-degeneracy assumptions, but these are not made concrete enough for people to judge when the guarantees apply. And the claim that forgetting is primarily driven by radial/time-like distortion s not directly validated quantitatively. The main evidence consists of an ablation and a small qualitative Flickr30K traversal case study, which is insufficient to support that claim to the strength stated in your paper.

3. HCML or HMCL? There are several inconsistencies in the paper. There is also an appendix cross-reference error: the paper says dataset stats are in Appendix G.2, but the appendix shows G.1. Dataset Details and Task Stream. Please proofread it.

---

> ### Author Rebuttal · Authors · 2026-03-31
>
> We sincerely thank the reviewer for the constructive comments and positive feedback. Below, we use **W/Q** for weaknesses/questions and **R** for our responses.
>
> **W1 & Q1: The gap between the theory and the algorithm.**
>
> **R:** Thanks for the constructive comment.  The implementation–theory gap is narrower than it may appear, because the practical update is directly constructed from the canonical admissible direction in Corollary 2. At stage $t$, let $P_{t-1}$ be the projector onto the old-task subspace, and let $\delta_t$ be the raw update directon. HMCL first projects it as $g_t:=(I-P_{t-1}),$ so $g_tP_{t-1}=0$ by construction. Let $M_t=\beta_tM_{t-1} + (1-\beta_1)g_t$ denote Adam’s first-moment estimate, with $M_0=0$. Since $P_{t-1}$ is fixed within the stage, it follows that $M_tP_{t-1}=\beta_1M_{t-1}P_{t-1}$, and hence, by induction,  $M_tP_{t-1} =0$ for all optimization steps. Therefore, momentum accumulation itself remains exactly in the admissible subspace and does not create additional leakage across multiple epochs.
>
> Our implementation uses `weight-decay = 0`. Let $U_t$ be the actual parameter update at step $t$, and $D_t$ the coordinate-wise preconditioner from Adam’s second-moment estimate. Then, $U_t=-\eta(D_t\odot M_t)$, and its leakage into the protected old-task subspace satisfies $\|U_tP_{t-1}\|_F\leq \eta_t \epsilon_t\|M_t\|_F, \epsilon:=\min||D_t-c1||$. Here $c$ is the best scalar approximation to $D_t$, so $\epsilon_t$ measures the anisotropy of the preconditioner. Hence, the optimizer mismatch is controlled only by the step size and its anisotropy term: it vanishes if $D_t$ is isotropic. At the representation level, the remaining discrepancy is only the standard $O(\eta_t^2)$ from Proposition 1. Thus, the implemented optimizer is an approximate realization of the canonical admissible update. In practice, we also conducted additional experiments with AdamW under **nonzero weight decay**, and our method still showed strong effectiveness, reducing forgetting by **70%** ([Anonymous Table](https://anonymous.4open.science/w/hcmls-EC11/RebuttalFigurePageGithub/?space=space-c&item=table-gesz-w1-q1-hmcl-mr)). This further suggests that the resulting optimizer-induced error remains well controlled in practice.
>
> **W2.1: Non-degeneracy assumption.**
>
> **R:** Thanks, and we agree that the non-degeneracy assumption should be stated more explicitly. Non-degeneracy assumption is not a restrictive special case, but the regime in which preservation is most tightly identifiable. In the non-degenerate case, old-task geometry is fully informative, so the admissible preservation map is most strongly constrained, yielding the sharpest characterization. Importantly, our method is not limited to this regime: when some directions collapse, the geometric constraints become less informative, so the preservation map is less uniquely determined, but the same framework still applies without modification. Thus, the assumption serves to characterize the strongest case, while **the method itself remains broadly applicable across all data regimes in practice.**
>
> **W2.2 & Q2: Quantitative forgetting analysis.**
>
> **R:** Thanks very much, and we apologize for the ambiguity caused by our wording in the abstract and conclusion. We did not intend to suggest that forgetting is solely caused by radial distortion. Rather, forgetting in hyperbolic continual learning includes both semantic forgetting (P1 & P2: degradation of intra-/inter-modal similarity structure) and hierarchy-related forgetting (P3: degradation of semantic specificity and partial order). The radial dimension is particularly tied to the latter (as we claimed in Section 4.2). Our method is **derived and designed to preserve all three properties** (P1/P2/P3).
>
> To address this concern, we add a quantitative dataset-wise drift analysis in [Anonymous Figure](https://anonymous.4open.science/w/hcmls-EC11/RebuttalFigurePageGithub/?space=space-c&item=figure-gesz-w2-2-q2-quantitatively-forgetting-analysis) that measures four complementary forms of forgetting-related representation drift: `radial drift` (hyperbolic radius / hierarchy), `angular drift` (within-modality directional structure), `cross-modal drift` (image-text relational structure), and `paired distance drift` (distance increase between matched image-text pairs). Averaged over the 14 old tasks, HMCL reduces these four drifts, which are significantly reduced by 61.0%, 57.8%, 73.8%, and 59.5%, respectively (p ≤ 3.1e-4). This provides direct quantitative evidence that HMCL mitigates both hierarchy and relation-related distortions associated with forgetting.
>
> **W3: Typos.**
>
> **R:** Thanks. We will fix all typos, and add a limitation section.
>
> **Q3: Stability of the task orders.**
>
> **R:** We randomly shuffled the task order and found that our method still performs well ([Anonymous Table](https://anonymous.4open.science/w/hcmls-EC11/RebuttalFigurePageGithub/?space=space-c&item=table-gesz-task-order-stability)).

---

> > ### Author Rebuttal · Reviewer_GesZ · 2026-04-02
> >
> > Thank you for the rebuttal. My concerns have been addressed. Please make sure you add all anonymous tables to your paper/appendix. I will improve my score and good luck!

---

> > > ### Author Response · Authors · 2026-04-02
> > >
> > > Thank you very much for your kind acknowledgement and encouraging comments. We are glad that our rebuttal has addressed your concerns. We will make sure to add all anonymous tables to the paper/appendix in the final version to further improve its completeness and presentation. Thank you again for your valuable time and suggestions!

---

### Official Review · Reviewer_wj2p · 2026-03-12

**Soundness:** 3
**Presentation:** 3
**Significance:** 3
**Originality:** 3
**Overall Recommendation:** 4
**Confidence:** 3

**Summary:**

The paper studies a central problem in continual learning for hyperbolic multimodal representations. It is motivated by the observation that hyperbolic semantic structure is encoded through geometric quantities, and continual parameter updates that violate the underlying geometric invariants can distort semantic similarity and hierarchical relationships, ultimately leading to catastrophic forgetting.

The paper makes the following contributions:

Theoretical characterization. The authors provide a theoretical analysis of representation preservation in hyperbolic continual learning, showing that preserving semantic structure requires a shared hyperbolic isometry across modalities.

First-order forgetting analysis. A first-order analysis suggests that forgetting is primarily driven by distortions in the radial dimension, which encodes the semantic hierarchy in hyperbolic space.

Training method. Based on this theoretical insight, the authors propose a training method, HMCL, which restricts parameter updates to geometry-preserving directions derived from the hyperbolic structure.

**Compliance With Llm Reviewing Policy:**

Affirmed.

**Final Justification:**

My concerns have been addressed, and I maintain my score at 4.

**Key Questions For Authors:**

Please refer to Weakness 1, 2 and 3.

**Limitations:**

No. Discussions of limits in Weakness 1 and 2 are recommended.

**Strengths And Weaknesses:**

Strength:

1. Well-defined theoretical formulation

The paper explicitly formulates the stability requirements for hyperbolic continual learning through three preservation conditions: intra-modal preservation, inter-modal preservation, and hierarchical preservation. These conditions are expressed in terms of the invariance of Lorentzian inner products and spatial norms, providing a clear geometric characterization of representation stability.

2. Interpretable theoretical insights

The proposed theory leads to several intuitive interpretations. In particular, the radial dimension corresponds to semantic hierarchy, while hyperbolic isometries correspond to knowledge preservation. This connection between hyperbolic geometry and catastrophic forgetting provides meaningful insight into the mechanisms underlying representation degradation in continual learning.

3. The method is theoretically motivated

The proposed update rule is derived directly from geometric constraints, rather than heuristic design choices. Specifically, the update procedure (1) projects gradients onto a subspace orthogonal to representations of previous tasks, and (2) preserves hierarchical and similarity relations in hyperbolic space. This derivation provides a strong conceptual justification for the proposed algorithm.

Weakness:

1. Theoretical assumptions are not sufficiently justified

The main theorem relies on a non-degeneracy condition on the learned representations. However, the paper does not clearly discuss (1) under what conditions this assumption holds in practice, (2) whether real multimodal embeddings satisfy it, or (3) how violations of this assumption would affect the proposed method. The lack of such discussion weakens the practical relevance of the theoretical guarantees.

2. The analysis is limited to first-order updates

The hierarchy-preservation analysis is based on first-order approximations of parameter updates. In practice, however, model training involves finite learning rates and nonlinear optimization dynamics. It remains unclear whether the theoretical guarantees remain valid beyond infinitesimal update regimes, which raises questions about their applicability to real training scenarios.

3. Potential oversimplification of forgetting mechanisms

The paper argues that catastrophic forgetting in hyperbolic continual learning is primarily caused by distortions in the radial dimension. However, forgetting may also arise from other factors, such as changes in angular relations or interactions between modalities. The paper does not provide empirical evidence verifying that radial distortions are the dominant source of forgetting, which leaves this claim insufficiently supported.

---

> ### Author Rebuttal · Authors · 2026-03-31
>
> We sincerely thank the reviewer for the constructive comments and positive feedback. Below, we use **W/Q** for weaknesses/questions and **R** for our responses.
>
> **W1:** **Regarding the non-degeneracy assumption.**
>
> **R:** Thanks for the comment, and we agree that the role of the non-degeneracy assumption should be stated more explicitly. We would clarify that the non-degeneracy assumption is not a restrictive special case, but the regime in which preservation is most tightly identifiable. In the non-degenerate case, old-task geometry is fully informative, so the admissible preservation map is most strongly constrained, yielding the sharpest characterization. Importantly, our method is not limited to this regime: when some directions collapse, the geometric constraints become less informative, so the preservation map is less uniquely determined, but the same framework still applies without modification. Thus, the assumption serves to characterize the strongest case, while **the method itself remains broadly applicable across all data regimes in practice.**
>
> **W2: The analysis is limited to first-order updates.**
>
> **R:** Thanks for raising this important point. We agree that the hierarchy-preservation analysis in the main text is first-order. Its purpose is to characterize the local geometric condition under which continual updates preserve the hyperbolic hierarchy, rather than to claim an exact global guarantee for arbitrary optimization trajectories.
>
> That said, the result is not merely infinitesimal. Corollary 1 identifies the radial component of the update as the leading source of hierarchy distortion, and our admissible update removes this component at first order. Therefore, under a finite update, we do not claim exact preservation of (P3); instead, the deviation from (P3) has no first-order term and only appears through higher-order effects. More concretely, if we write $\Delta z=z_{t-1}^{m,t} - z_{t-1}^{m,t}$, then under the admissibility condition, the hierarchy-related radial deviation satisfies
> $||z_t^{m,t}[1:d]|| - ||z_{t-1}^{m,t-1}[1:d]||=O(||\Delta z||^2)$.
> Since $\Delta z$ is induced by parameter update $W_{m,t} = W_{m,t-1} - \eta \Delta W_m$, and the representation map is locally smooth, we have $||z||=O(\eta)$, which yields an $O(\eta^2)$ hierarchy error under finite learning rates.
>
> In this sense, our theory remains relevant beyond the infinitesimal regime: it identifies and suppresses the dominant first-order hierarchy drift, which explains why the method is substantially more stable than unconstrained continual updates. We will clarify this distinction between exact preservation and first-order preservation in the revision.
>
> **W3: Potential oversimplification of forgetting mechanisms.**
>
> **R:** Thanks very much for pointing this out, and we apologize for the ambiguity caused by our wording in the abstract and conclusion. We did not intend to suggest that forgetting is solely caused by radial distortion. **Rather, forgetting in hyperbolic continual learning includes both semantic forgetting** (P1 & P2: degradation of intra-/inter-modal similarity structure) and **hierarchy-related forgetting** (P3: degradation of semantic specificity and partial order). The radial dimension is particularly tied to the latter (as we claimed in Section 4.2). **Our method is derived and designed to preserve all three properties** (P1/P2/P3), and thus addresses both semantic and hierarchical distortion. We will clarify this point in the revised abstract and conclusion.
>
> To address this concern, we add a quantitative dataset-wise drift analysis (see [**Anonymous Figure**](https://anonymous.4open.science/w/hcmls-EC11/RebuttalFigurePageGithub/?space=space-b)) that directly measures multiple forms of geometric change associated with forgetting. In particular, `radial drift` captures changes in the hyperbolic radius of old-task embeddings, reflecting hierarchy / time-like distortion; `angular drift` captures changes in within-modality directional structure after factoring out magnitude; `cross-modal drift` captures changes in the global relational structure between image and text embeddings; and `paired distance drift` captures how much the geodesic distance between matched image-text pairs changes over time. These results provide **direct quantitative evidence that forgetting is accompanied by both hierarchy-related and relation-related geometric drift.** **Importantly, our method significantly reduces the drift corresponding to P1, P2, and P3, which explains its stronger forgetting mitigation effect.**
>
> **Limitation.** We will add a section to discuss the limitations of the method to guide future exploration.

---

> > ### Author Rebuttal · Reviewer_wj2p · 2026-04-03
> >
> > Thanks for the rebuttal, which has addressed my concerns. I would like to keep my original rating.

---

> > > ### Author Response · Authors · 2026-04-03
> > >
> > > We sincerely appreciate your positive feedback and are pleased that our rebuttal has addressed your concerns. We will incorporate the rebuttal into the final version of our manuscript.

---

### Official Review · Reviewer_R1as · 2026-03-13

**Soundness:** 3
**Presentation:** 2
**Significance:** 3
**Originality:** 3
**Overall Recommendation:** 4
**Confidence:** 3

**Summary:**

The paper tackles the problem of catastrophic forgetting in continual learning,specifically for multimodal models that use hyperbolic representation space.They show that when learning new tasks,the hyperbolic structure (like the semantic hierarchy of concepts) gets destroyed if updated naively.To fix this,they use geometric analysis to figure out what kind of parameter updates keep the old structure safe.They propose a method called HMCL that restricts parameter updates to directions that don't mess up the old representations by using a projection matrix.They tested it on models like MERU and HyCoCLIP and showed it forgets much less than traditional baselines like EWC or GEM.

**Compliance With Llm Reviewing Policy:**

Affirmed.

**Key Questions For Authors:**

-In Algorithm 1,you use SVD/PCA to get the basis V from old-task representations. Does the size of this matrix and the memory needed grow linearly as we see more and more tasks?Will it eventually run out of memory if the task sequence is very long?

**Limitations:**

No,not really.The Impact Statement just says there are no specific societal consequences to highlight.I think the authors should explicitly add a discussion about their technical limitations.

**Strengths And Weaknesses:**

Strengths
1）Bringing continual learning into hyperbolic multimodal representation space is genuinely fresh. Most CL work assumes Euclidean geometry, so the non-Euclidean angle is a meaningful new perspective.
2）Results look strong: clear gains over vanilla fine-tuning and older regularization methods (e.g., EWC). The qualitative traversal study in Fig. 4 helps show hierarchy preservation, not just metric improvements.

Weaknesses
1) Sections 4–5 are mathematically dense and hard to audit; some key steps could use more unpacking or sanity checks. Also, training only a lightweight linear layer while freezing backbones feels restrictive for continual learning; it’s unclear whether this reflects standard hyperbolic practice or a convenience setting.
2) Euclidean CL baselines (EWC, GEM) are “ported” by swapping in a hyperbolic loss. It’s not obvious this is a fair or strong adaptation, since these methods weren’t designed for hyperbolic geometry; the comparison may understate what a geometry-aware baseline could achieve.

---

> ### Author Rebuttal · Authors · 2026-03-31
>
> We sincerely thank the reviewer for the constructive comments and positive feedback. Below, we use **W/Q** for weaknesses/questions and **R** for our responses.
>
> **W1.1:** **The paper is hard to audit.**
>
> **R:** Thank you for the helpful suggestion. Due to space limitations, we only included a brief proof sketch in the main text. In the revision, we will improve the presentation of Sections 4–5 by providing a more detailed proof sketch in the appendix, together with clearer intermediate explanations, to make the derivations easier to audit. We will also add sanity-check diagnostics to verify that the proposed update behaves as intended.
>
> **W1.2: The training setting seems overly restrictive.**
>
> **R:** Thanks for the comment. Using pretrained encoders is a deliberate and practically motivated design choice for our method, rather than a convenience-only simplification. Our goal is to study parameter-efficient continual adaptation in a knowledge-preserving regime: freezing the backbone reduces training cost and limits interference with previously acquired representations, while the lightweight trainable layer makes the effect of the proposed hyperbolic adaptation mechanism easier to isolate and analyze.
>
> From a theoretical perspective, restricting updates to a small module reduces interference with previously acquired representations, and our projected admissible update further cancels the first-order drift of old-task representations, which makes this regime favorable to knowledge preservation (Section 4.2, Appendix F). At the same time, the learnable capacity for new tasks comes from the remaining admissible degrees of freedom outside the preserved old-task subspace; *in high-dimensional pretrained representation spaces, this remaining adaptation space is substantial in practice.* Empirically, we find that such a lightweight layer already provides enough flexibility to absorb new task information while better retaining prior knowledge (Section 5.2).
>
> **W2:  Fairness of baselines.**
>
> **R:** Thank you for this important comment. **To the best of our knowledge, there is currently no continual learning method specifically designed for hyperbolic multimodal representations.** Therefore, we are unable to compare against an established geometry-aware baseline in this setting. We thus adopt what we believe is the fairest comparison protocol: we adapt representative continual learning baselines to the same hyperbolic objective, while keeping their original continual-learning mechanisms unchanged. **This ensures that all methods are evaluated under the same representation geometry, and the remaining gap reflects the continual-learning strategy itself.** The reason these baselines still underperform is that, although they are trained with a hyperbolic loss, their update rules remain geometry-agnostic and do not explicitly preserve the hyperbolic invariants that matter, such as shared cross-modal isometry and hierarchy preservation. Importantly, making these methods truly geometry-aware is not straightforward, since it requires update rules with explicit geometric invariance guarantees, which is exactly what motivates our derivation. We will clarify this point in the revision. We will clarify this point in the revision.
>
> **Q1: Scalability for long task sequences.**
>
> **R:** Thank you for this important question. Our method does not maintain a separate basis for each previous task, nor does it store historical representations. Instead, it only keeps a running summary of the retained old-task subspace through a fixed $d\times d$ covariance matrix, which is used to construct the projection in Algorithm 1. As a result, the memory overhead is constant with respect to the number of tasks, rather than increasing linearly with the number of tasks, which makes the method scalable to long task sequences.
>
> **Limitation.** Thanks for the suggestion.  We will add a section to discuss the limitations of the method to guide future exploration.

---

### Decision · Program_Chairs · 2026-04-30

**Decision:**

Accept (regular)

**Comment:**

This paper studies continual learning in hyperbolic multimodal representation spaces and proposes a geometry-driven framework to mitigate catastrophic forgetting. Reviewers agree the problem is novel and meaningful, extending continual learning beyond Euclidean settings into hyperbolic geometry. Theoretical contributions are clear and well-motivated, with preservation conditions and hyperbolic isometries providing an interpretable view of representation stability, and the update rule derived directly from geometric constraints. Empirically, the method consistently improves over strong baselines across multiple backbones, with qualitative results supporting hierarchy preservation. Concerns are minor and are largely addressed in rebuttal. Overall, there is a consensus on the paper’s originality, soundness, and impact, supporting acceptance.